# Abrasive Wear Resistance of Plasma-Nitrided Ti Enhanced by Ultrasonic Surface Rolling Processing Pre-Treatment

**DOI:** 10.3390/ma12193260

**Published:** 2019-10-06

**Authors:** Dingshun She, Shihao Liu, Jiajie Kang, Wen Yue, Lina Zhu, Chengbiao Wang, Haidou Wang, Guozheng Ma, Li Zhong

**Affiliations:** 1School of Engineering and Technology, China University of Geosciences (Beijing), Beijing 100083, China; shedingshun@163.com (D.S.); cugbyw@163.com (W.Y.); zhulina@cugb.edu.cn (L.Z.); wanghaidou@aliyun.com (H.W.); 2Zhengzhou Institute, China University of Geosciences (Beijing), Zhengzhou 451283, China; cbwang@cugb.edu.cn; 3Zhengzhou Institute of Multipurpose Utilization of Mineral Resources, Chinese Academy of Geological Sciences, Zhengzhou 450006, China; 4National Key Lab for Remanufacturing, Army Academy of Armored Forces, Beijing 100072, China; magz0929@163.com; 5SEU-FEI Nano-Pico Center, Key Laboratory of MEMS of Ministry of Education, Southeast University, Nanjing 2100096, China

**Keywords:** titanium, plasma nitriding, surface nano-crystallization, lunar regolith particles, abrasive wear

## Abstract

The objective of the given work was to investigate abrasive wear behaviours of titanium (Ti) treated by ultrasonic surface rolling processing (USRP) pre-treatment and plasma nitriding (PN). Simulated lunar regolith particles (SLRPs) were employed as abrasive materials during characterization of tribological performances. The experimental results showed that SLRPs cause severe abrasive wear on Ti plasma-nitrided at 750 °C via the mechanism of micro-cutting. Due to the formation of a harder and thicker nitriding layer, the abrasive wear resistance of the Ti plasma-nitrided at 850 °C was enhanced, and its wear mechanism was mainly fatigue. USRP pre-treatment was effective at enhancing the abrasive wear resistance of plasma-nitrided Ti, due to the enhancement of the hardness and thickness of the nitride layer. Nevertheless, SLRPs significantly decreased the friction coefficient of Ti treated by USRP pre-treatment and PN, because the rolling of small granular abrasives impeded the adhesion of the worn surface. Furthermore, USRP pre-treatment also caused the formation of a dimpled surface with a large number of micropores which can hold wear debris during tribo-tests, and finally, polishing and rolling the wear debris resulted in a low friction coefficient (about 0.5).

## 1. Introduction

For lunar exploration, micro-/nano-scaled lunar regolith particles that cover the surface of the moon always cause serious abrasive wear of a lunar rover [1,2,3,4,5]. A great number of lunar rover parts, such as the drill pipe and sleeve used on the lunar drilling, are made of titanium (Ti) and its alloys due to its low density, high strength, excellent corrosion resistance, etc. [4,6,7,8,9,10]. Nevertheless, the poor abrasion wear resistance of Ti and its alloys restricts the service life and operating limit of drill devices [8,10].

Thus far, a great number of surface treatments, for insistence, physical vapor deposition (PVD), chemical vapor deposition (CVD), surface texturing, laser deposition, surface alloying and ion implantation, and surface thermo-chemical treatment, have been introduced to improve the tribological properties of Ti and its alloy [8,10,11,12,13,14,15]. However, among these surface treatments, many are limited by poor film–substrate cohesion, low processing efficiency and/or vast production cost. Accordingly, it is essential to find a surface treatment with lower cost and higher efficiency, as well as non-pollution, to enhance the tribological performances of Ti drilling devices.

Plasma nitriding, a potential method for enhancing the service life of Ti drilling devices, can produce a gradient nitrided layer with a thickness of ~10^2^ μm and a hardness higher than 1000 HV [7,9,10]. Especially in combination with surface nano-crystallization pre-treatment, nitriding can exhibit a higher nitriding efficiency under a lower nitriding temperature [7,16,17,18,19,20]. Ge et al. [18] revealed that surface nano-crystallization pre-treatment can produce a thicker and harder nitrided layer on Ti6Al4V alloy compared with direct gas nitriding. Korshunov et al. [19] demonstrated that severe plastic deformation pre-treatment can improve wear resistance of gas-nitrided Ti. She et al. [7] proved that ultrasonic surface rolling processing (USRP) pre-treatment, a kind of efficient surface nano-crystallization, effectively decreases the optimized nitriding temperature for improving the vacuum tribological performance of plasma nitriding. Nevertheless, the abrasive wear behaviours of plasma-nitrided Ti pre-treated by USRP are still far from being understood.

Here, simulated lunar regolith particles (SLRPs) were employed as abrasive materials to investigate the effects of abrasive materials on the tribological performances of Ti, and USRP pre-treatment was adopted as a surface nano-crystallization pre-treatment of plasma nitriding to enhance the abrasive wear resistance of Ti. Furthermore, the effects of USRP pre-treatment on the abrasive wear behaviours of plasma-nitrided Ti was analysed in detail. In this paper, a novel surface modification method used to enhance abrasive wear resistance of lunar drilling devices is introduced.

## 2. Experimental Details

### 2.1. Materials and Surface Treatments

Annealed commercial titanium TA2 (ASTM Gr.3) bar made by Baoji Titanium Industry Co., Ltd. was cut into slices of 25 × 25 × 3 mm via the high-pressure water jet technique. The chemical composition of TA2 slices provided by the company is listed in Table 1. After cutting, TA2 slices were mechanically ground and polished to obtain a mirror surface, and then ultrasonically cleaned in ethanol for 20 min.

The prepared TA2 slices were first treated by USRP pre-treatment. During USRP treatment, the ultrasonic vibratory signal from the IL-10 model generator was converted into mechanical vibration via a USP-125 model transducer booster with a cemented carbide plunger chip. This resulted in severe plastic deformation on the surface of the TA2 slices. The transducer booster worked like a lathe tool, and Ti slices were fixed on the chuck. The detailed principles and processes of USRP treatment can be found in [20]. The parameters of USRP pre-treatment are shown in Table 2.

After USRP pre-treatment, the TA2 slices were plasma-nitrided using an LDM-100 model furnace. TA2 slices surrounded by an active screen were placed on the cathode, and the furnace wall acted as an anode. The TA2 slices were first cleaned by sputtering under an ammonia (NH_3_) atmosphere. Prior to nitriding, NH_3_ was extracted out, and N_2_ then filled the vacuum chamber. Plasma nitriding was performed at 750 and 850 °C for 8 h (optimized temperatures for enhancing vacuum tribological performances of Ti [7,9]). Table 3 shows the detailed plasma nitriding parameters.

### 2.2. Micro-Structure and Micro-Hardness Characterization

The phase composition of the Ti slices and simulated lunar regolith particles were investigated by a D/max-2500X-ray model diffractor (XRD) with a Cu–Kα radiation source (wavelength, 0.15406 nm).

The surface morphology of the Ti slices and simulated lunar regolith particles were observed by ZEISS MERLIN Compact scanning electron microscope (SEM) (Carl Zeiss AG, Jena, Germany), and its appendant energy disperse X-ray spectroscopy (EDS) was used to measure the elemental composition. The surface morphology was observed under a secondary electron image system. During EDS elemental composition measurements, the accelerating voltage was 15 kV, and the working distance was about 15 mm.

A Nano-Map-D model 3D profiler was adopted to characterize the 3D topographies and surface roughness of Ti. Using an optical microscope, the cross-sectional morphology of Ti was investigated in detail via metallographic observation.

The hardness measurement was tested by the HVS-1000Z model hard-meter under the load of 50 g and the duration of 10 s, according to the ASTM standard E384-11e1.

### 2.3. Tribolo-Tests

To characterize abrasive wear resistance of lunar drilling devices, homogeneous simulated lunar regolith particles without agglomeration (Figure 1a) were chosen as the abrasive medium. The particle size of the original lunar soil was micro-/nano-sized [4,5,21,22,23]; therefore, the distribution of simulated lunar regolith particles also showed a Gaussian distribution centered around 80 μm (Figure 1b). According to the characteristics of lunar mare basalts, simulated lunar regolith particles composed of SiO_2_, Al_2_O_3_, CaO, TiO_2_, FeO and MgO were made from volcanic rock [22,23].

Tribological tests were performed on a MS-T3000 ball-on-disk model tribo-meter produced by Lanzhou, Gansu Province, China, Huahui instrument technology Co., Ltd. During the tribo-tests, the Ti slices were rotated, driven by a motor at room temperature (28 ± 0.5 °C), and a SiO_2_ ball with a diameter of 6.0 mm was fixed as the counterpart. Simulated lunar regolith particles acted as abrasives evenly distributed on the surface of the disc. A cylindrical cover was used to prevent the particles from being thrown out. The normal load was 3 N with a maximum contact pressure of 0.50 GPa (Hertzian contact), and the duration was 30 minutes. The friction force was measured every 1.5 s, which was finally converted to an output friction coefficient. The rotating speed was 200 r/min with a linear velocity of 0.046 m/s. The diameter of the wear track was 2.2 mm, and the total sliding distance of the tests was about 83.02 m. Each of the tribological tests was carried out three times to ensure the reliability and repeatability of the results. After the tribo-tests, the wear volumes of the Ti slices measured by a 3D profiler were converted to a wear rate (K) according to the Archard wear equation:(1)$$K=\frac{V}{F\times S}$$
where V is the wear volume, F is the normal load, S is the sliding distance, and K is the wear rate per unit load and per unit distance [7,24].

The wear volume of the counter-ball was shown via the diameter of the wear scar, which can be converted to the wear rate via a calculation. The wear debris and worn surface were observed by SEM and its appendant EDS to reveal the wear mechanism of plasma-nitrided Ti.

## 3. Results and Discussion

### 3.1. Micro-Structure and Morphology

XRD diffraction profiles of Ti treated by USRP pre-treatment and plasma nitriding are shown in Figure 2. The diffraction peaks of the a-Ti phase for the USRP Ti are broader and weaker than that of the untreated Ti, which is attributed to grain refinement and concentration of defects. It has also been demonstrated in our pervious works [6,7] that USRP can result in the formation of a nano-crystal surface with an average size of 38 nm. After plasma nitriding at 750 and 850 °C, the USRP pre-treated Ti (S&N750 and S&N850) with a nano-crystal surface showed high-insensitive diffraction peaks of nitrides including α-Ti(N), ε-Ti_2_N and δ-TiN, compared with the Ti that was directly plasma-nitrided (N750 and N850). This result was attributed to the formation of a grain boundary and defects that can promote the formation of nitride [16,17,25].

Figure 3 shows typical SEM surface morphologies of untreated and USRP Ti plasma-nitrided at 750 (N750 and S&N750) and 850 °C (N850 and S&N850). It can be seen that the nitride phases formed on the untreated and plasma-nitrided and USRP-treated Ti are shown as a great number of nano-particles. The S&N750 Ti and S&N850 Ti show a compact and dimpled surface, but the N750 and N850 Ti show a loosely-packed surface with some micro-cracks. Obviously, USRP pre-treatment results in the formation of nano-sized nitride particles.

Figure 4 shows 3D profiler images of the untreated and USRP pre-treated Ti plasma-nitrided at 750 and 850 °C. After USRP pre-treatment, some micro-grooves formed on the Ti surface. The USRP pre-treated Ti showed a rougher and dimpled surface comparing with directly plasma-nitrided Ti. The formation of a dimpled surface is attributed to the coaction of micro-grooves caused by USRP pre-treatment (Figure 4d) and cathode sputtering during plasma nitriding [26].

Typical cross-sectional micro-photographs of the Ti treated by USRP pre-treatment and plasma nitriding are shown in Figure 5. A gradient deformation layer with ultra-fine grain can be observed after USRP treatment. Evidently, USRP pre-treatment is effective in enhancing the thickness of the nitride layer compared with the Ti directly treated by plasma nitriding. Based on previously published results [7,16,17,20,25], it has been proposed that this thickness enhancement is due to defects in the deformation layer caused by USRP pre-treatment acting as channels for atom diffusion during nitriding. Nonetheless, the ultra-fine grain prepared via USRP also resulted in a larger grain-sized surface on plasma-nitrided Ti, resulting from secondary recrystallization.

### 3.2. Micro-Hardness

Figure 6 shows the micro-hardness and thickness of plasma-nitrided Ti pre-treated by USRP. The micro-hardness is effectively enhanced via USRP pre-treatment and plasma nitriding, and it gradually decreases from the top surface to the substrate. The enhancement of hardness for USRP pre-treatment is attributed to the grain-refinement [27]; and that for plamsa-nitriding is due to the formation of the gradient nitride layer [20,25,26]. The S&N850 Ti showed a noticeable surface hardness of 1100 HV_0.05_. Notably, USRP pre-treatment enhanced the surface hardness and the thickness of the nitride layer on the Ti plasma-nitrided at either 750 or 850 °C.

### 3.3. Friction Coefficient

Friction coefficients of the Ti discs treated by USRP pre-treatment and plamsa nitriding sliding against a SiO_2_ ball with abrasive SLRPs are shown in Figure 7. It can be seen that abrasive SLRPs can result in a low friction coefficient, compared with the condition without abrasive SLRPs. Moreover, the USRP-treated Ti showed a low friction coefficient, compared with the untreated Ti sliding either with or without abrasive SLRPs, prior to plasma nitriding. Interestingly, the friction coefficients of the Ti treated by USRP pre-treatment and directly plasma-nitrided Ti, sliding against the SiO_2_ ball without SLRPs, are nearly consistent (about 1.28); nonetheless, USRP pre-treatment evidently also causes a low friction coefficient for plasma-nitrided Ti sliding with SLRPs. Average friction coefficients of S&N750 and S&N 850 Ti sliding against SiO_2_ ball with SLRPs are about 0.5, which are lower than that of the N750 (0.65) and N850 (0.7) Ti.

### 3.4. Wear Behaviours

Typical 3D worn surface morphologies of the Ti treated by USRP pre-treatment and plasma nitriding are shown in Figure 8. Apparently, the plasma-nitrided Ti sliding without abrasive SLRPs showed a mild wear with a smooth surface; whereas that sliding with abrasive SLRPs were seriously worn with obvious wear scars. Micro-grooves and peeling pits could be found on the worn surface of the N750 and S&N750 Ti. For the plasma-nitrided Ti sliding either with or without abrasive SLRPs, USRP pre-treatment resulted in a relatively mild wear.

The wear rate of the Ti treated by USRP pre-treatment and plasma nitriding is plotted in Figure 9. Evidently, abrasive SLRPs caused a higher wear rate of the Ti disc, compared with the ball directly sliding against the Ti disc. In addition, either USRP or plasma nitriding decreased the wear rate of the Ti. The S&N850 exhibited the best wear resistance among the untreated and pre-treated Ti that was plasma-nitrided at 750 and 850 °C.

Figure 10 shows optical microscope images of the surface morphologies of wear scars on the ball sliding against the Ti treated by USRP pre-treatment and plasma nitriding. Under the condition of sliding with abrasive SLRPs, the surfaces of the wear scar of the balls sliding against the N750 and S&N750 Ti (Figure 10e,f) showed some micro-plows and irregular edges, whereas that against the S&N850 Ti was relatively flat with a regular circular edge. The wear scar on the balls sliding against the N850 Ti with abrasive SLRPs (Figure 10g) contained two distinct surface morphologies: a bright one, shown by some micro-plows and irregular edges (like the balls sliding against the N750 and S&N750 Ti), and a dark one that was flat with a regular circular edge (like the balls sliding against the S&N850 Ti). When sliding without abrasive SLRPs, the wear scar of the balls sliding against the Ti treated by USRP pre-treatment and plasma nitriding (Figure 10a–d) were all present as nearly-flat circles. Futhermore, the wear scar of the balls sliding against the N750, S&N750 and N850 Ti with abrasive SLRPs were covered by some large bits of debris and granular wear debris. However, wear debris cover on the ball sliding against the S&N850 was only granular wear debris.

The wear scar diameter of the balls sliding against the Ti treated by USRP pre-treatment and plasma nitriding with abrasive SLRPs is plotted in Figure 11. Abrasive SLRPs are effective at increasing the wear of the ball. When sliding with abrasive SLRPs, the diameter of the wear scar on the ball sliding with the Ti nitrided at 750 °C was larger than that at 850 °C. The diameter of the wear scar on the ball sliding with the N850 Ti was much larger than that of the S&N850 Ti. The ball sliding against the S&N850 Ti showed the mildest wear compared with that against N750, S&N750 and N850 Ti. When sliding without abrasive SLRPs, the diameter of the wear scar on the ball sliding with the directly plasma-nitrided Ti and the Ti treated by USRP pre-treatment and plasma nitriding were consistent in size. It is worth noting that the abrasive SLRPs resulted in a low friction coeifficient but a high wear rate of the ball and Ti discs, which is in contrast to the law that a higher friction coefficient leads to a larger wear rate.

Figure 12 shows a typical SEM image of the surface morphology of the worn surface on the Ti treated by USRP pre-treatment and plasma nitriding. After sliding with abrasive SLRPs, micro-plowing and micro-scale granular wear debris could be observed on the worn surface of the untreated, USRP Ti and N750 Ti. According to the results of the EDS analyses shown in Table 4, it can be concluded that granular wear debris on a micro-scale was SLRPs. Therefore, the wear mechanism of the untreated, USRP Ti, N750 and S&N750 Ti is abrasive wear and micro-cutting. Similar to N750 Ti, the wear mechanism of the S&N750 Ti is the co-action of abrasive wear and micro-cutting. No SLRPs could be seen; however, the greatest amount of nano-sized wear debris was on the worn surface of the S&N750 Ti. This phenomenon resulted from that the high surface hardness of S&N750 Ti impeded micro-sized SLRPs press-in. On the worn surface of the N850 and S&N850 Ti, there was no wear debris but some spalling pits and micro-cracks were present, which demonstrates that wear is mainly due to fatigue.

After sliding without abrasive SLRPs, the worn surface of the untreated and USRP Ti (Figure 10a–d) featured micro-plowing and wear debris; and that of the N750 and S&N750 Ti was smooth. Some scaly wear debris covered the worn surface of the N850 and S&N850 Ti. These characteristics, seen in SEM images, indicates that serious micro-cutting occurred on worn surfaces of the untreated and USRP Ti; nonethless, the wear mechanism on the plasma-nitrided Ti was mild abrasive wear. According to the results of the EDS analyses shown in Table 4, the wear debris on the worn surface of the Ti were SiO_2_ and TiNx nano-particles. After plasma nitriding, the worn surface of the USRP pre-treated Ti showed more wear debris, compared with that of the untreated Ti. The aggregation of wear debris on the worn surface resulted from the higher hardness (Figure 6) caused by USRP pre-treatment, which can result in a higher wear rate of the ball (Figure 9 and Figure 11).

The hard abrasive SLRPs caused severe micro-cutting for the Ti with relatively low hardness, such as the untreated, USRP, N750 and S&N750 Ti. For the Ti plasma-nitrided at 850 °C with a high hardness, the anti-micro-cutting ability was enhanced. Accordingly, the N850 and S&N850 Ti showed a relatively low wear rate (Figure 9). The wear mechanism of the Ti plasma-nitrided at 850 °C is fatigue wear resulting from rolling of the abrasive SLRPs. In particular, the large-sized grain (Figure 5) caused by secondary recrystallization during plasma nitriding at a high temperature always causes poor bearing capacity and the resistance of crack propagation and growth [7,28]. Furthermore, abrasive SLRPs can impede the direct connections and adhesions between the ball and the Ti discs, and the co-action of polishing and rolling of abrasive SLRPs finally results in a low friction coefficient, compared with sliding without abrasive SLRPs. When sliding without abrasive SLRPs, the improvement of surface hardness via USRP and plasma nitriding was effective at enhancing the adhesive wear resistance of the Ti, according to the Archard relationship. Nevertheless, the adhesion between the ball and the Ti discs, and the high surface roughness, caused a friction coefficient higher than 1 (Figure 7). As a result, there was a high friction coefficient and low wear rate compared to sliding with abrasive SLRPs.

USRP pre-treatment enhances the hardness and thickness of the untreated and plasma-nitrided Ti (Figure 6). According to the Archard relationship [29], the improvement in hardness results in excellent adhesive wear resistance and anti-micro-cutting ability. As a result, the S&N750 Ti showed a lower wear rate than the 750 Ti when sliding with abrasive SLRPs. Meanwhile, the USRP Ti exhibited better wear resistance compared to the untreated Ti. For the N850 and S&N850 Ti, USRP pre-treatment resulted in a surface without micro-cracks, which caused an excellent fatigue wear resistance. Finally, the S&N850 Ti showed the lowest wear rate among the untreated and pre-treated Ti plasma-nitrided at 750 and 850 °C. Additionally, USRP pre-treatment resulted in a dimpled surface with a greater number of micropores on the surface of the plasma nitriding (Figure 3 and Figure 4). The dimpled surface could keep some wear debris on the worn surface (Figure 12), and finally, the rolling and polishing of abrasive SLRPs resulted in a low friction coefficient (Figure 7) [5,28,30]. Consequently, the S&N850 Ti showed the best friction reduction and anti-wear performance among the Ti treated by USRP pre-treatment and plasma nitriding when sliding with abrasive SLRPs.

## 4. Conclusions

(1) USRP pre-treatment enhances the abrasive wear resistance of plasma-nitrided Ti, due to the formation of a harder and thicker nitride layer compared with direct plasma nitriding.

(2) USRP pre-treatment decreases the friction coefficient of the plasma-nitrided Ti during abrasive wear tests, resulting from the formation of a dimpled surface.

(3) SLRPs impede the direct connection and adhesion; and hence, decrease the friction coefficient of (plasma-nitrided) Ti. Nevertheless, SLRPs also causes serious abrasive wear on the surface of (plasma-nitrided) Ti.

(4) The Ti treated by USRP pre-treatment and plasma nitriding at 850 °C showed optimized abrasive wear resistance and a friction coefficient of ~0.5.

## Figures and Tables

**Figure 1 materials-12-03260-f001:**
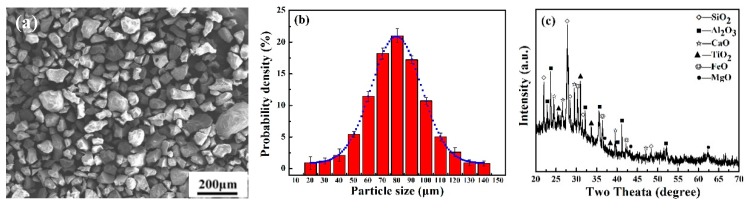
Microstructures of simulated lunar regolith particles: (**a**) morphology shown via secondary electron SEM images, (**b**) Gaussian distribution of particle size and (**c**) phase composition shown by XRD diffraction pattern.

**Figure 2 materials-12-03260-f002:**
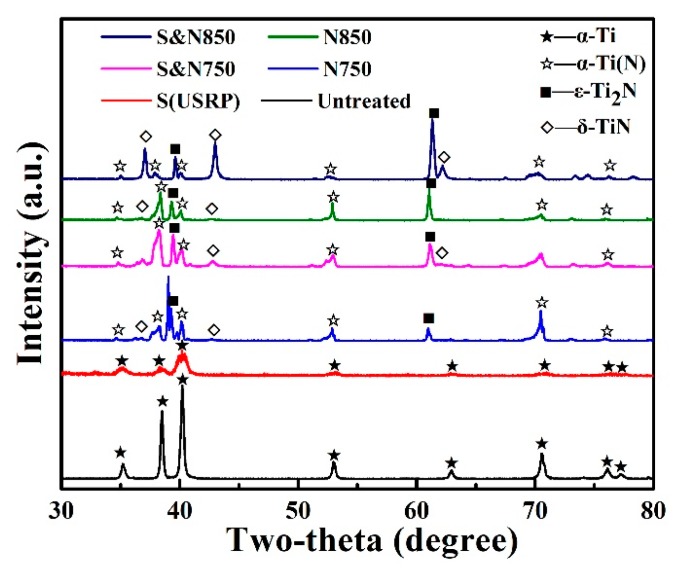
XRD diffraction profiles of Ti treated by USRP pre-treatment and plasma nitriding.

**Figure 3 materials-12-03260-f003:**
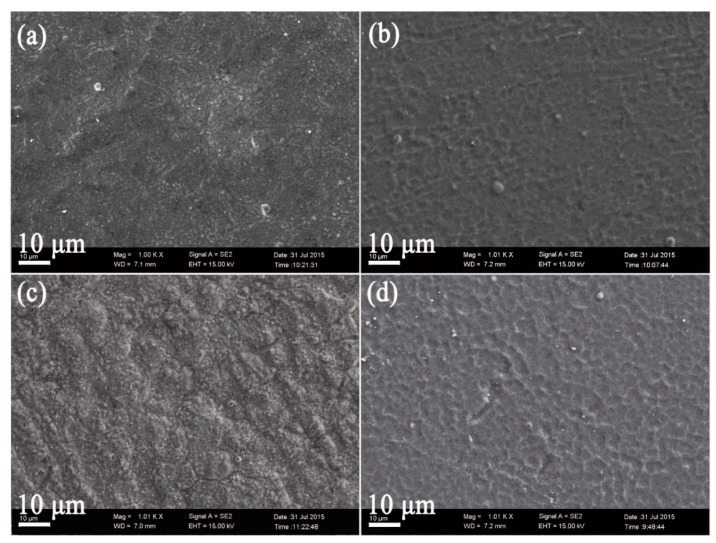
Typical SEM images showing surface morphologies of the (**a** and **c**) untreated and (**b** and **d**) USRP Ti plasma-nitrided at (**a** and **b**)750 °C and (**b** and **d**) 850 °C.

**Figure 4 materials-12-03260-f004:**
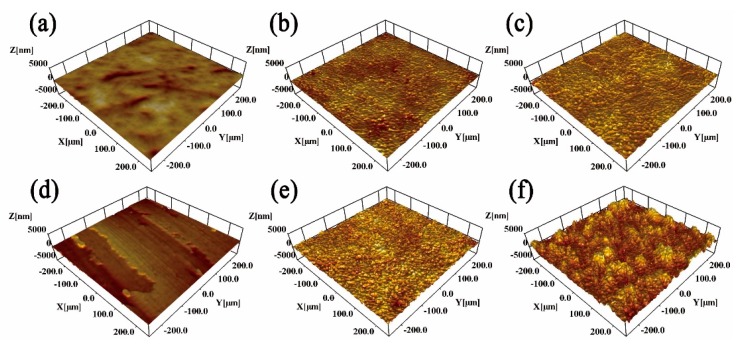
3D profiler images of the (**a**) Un-treated, (**b**) N750, (**c**) N850, (**d**) USRP, (**e**) S&N750 and (**f**) S&N850 Ti.

**Figure 5 materials-12-03260-f005:**
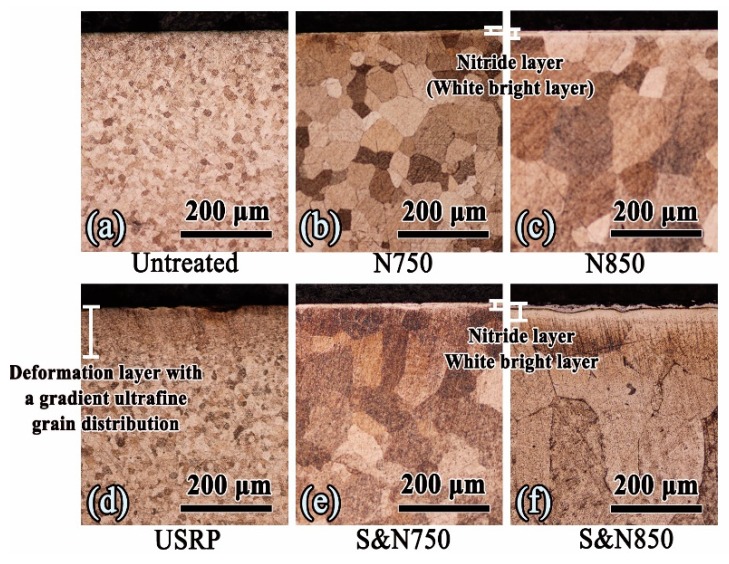
Typical cross-sectional micro-photographs of the (**a**) un-treated, (**b**) N750, (**c**)N850, (**d**) USRP, (**e**) S&N750 and (**f**) S&N850 Ti.

**Figure 6 materials-12-03260-f006:**
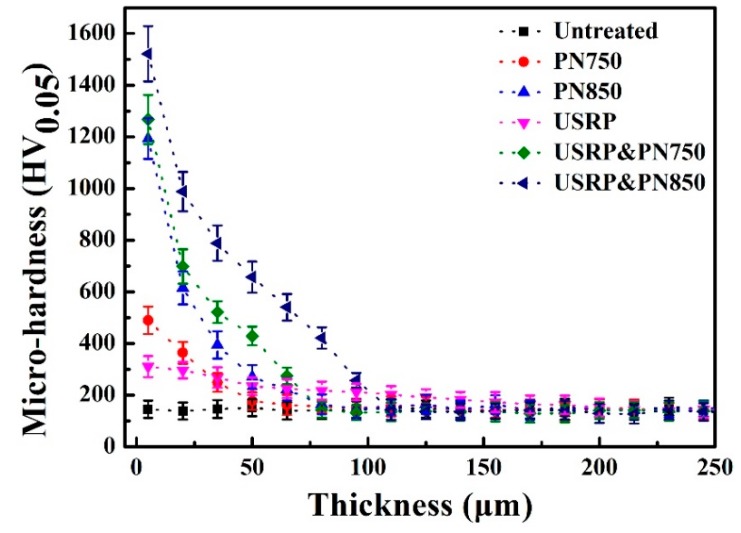
Micro-hardness of plasma-nitrided Ti pre-treated by USRP against the thickness.

**Figure 7 materials-12-03260-f007:**
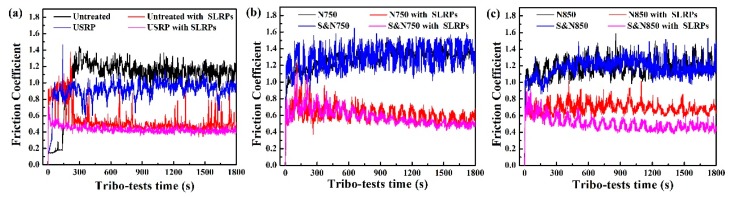
Friction coefficients of the (**a**) untreated and USRP pre-treated, (**b**) N750 and S&750 and (**c**) N850 and S&850 Ti sliding against a SiO_2_ ball with and without abrasive SLRPs.

**Figure 8 materials-12-03260-f008:**
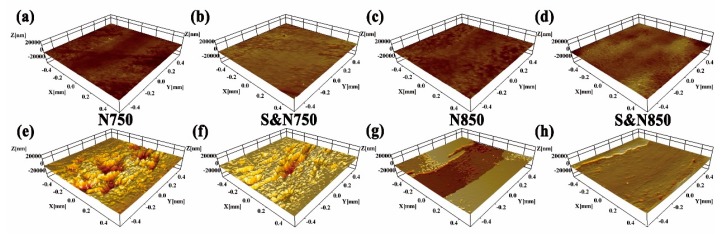
Typical 3D worn surface morphologies of the plasma-nitrided Ti sliding (**a**–**d**) without and (**e**–**h**) with abrasive SLRPs.

**Figure 9 materials-12-03260-f009:**
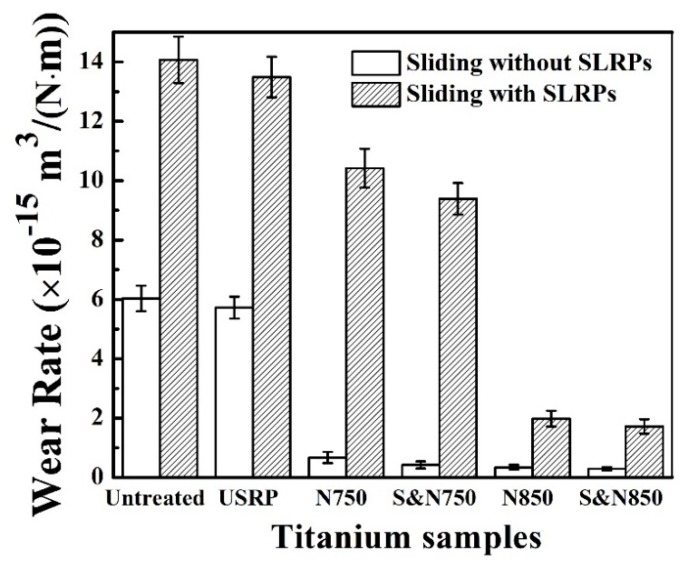
Wear rate of the Ti treated by USRP pre-treatment and plasma nitriding.

**Figure 10 materials-12-03260-f010:**
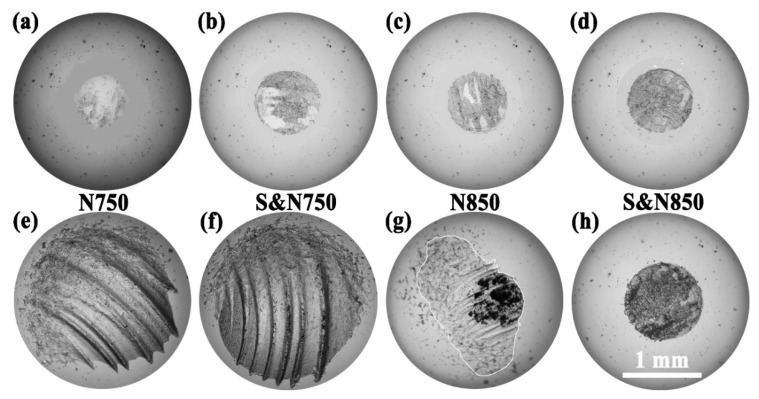
Optical microscope images showing the surface morphologies of the wear scar on the ball sliding against the Ti treated by USRP pre-treatment and plasma nitriding (**a**–**d**) without and (**e**–**h**) with abrasive SLRPs.

**Figure 11 materials-12-03260-f011:**
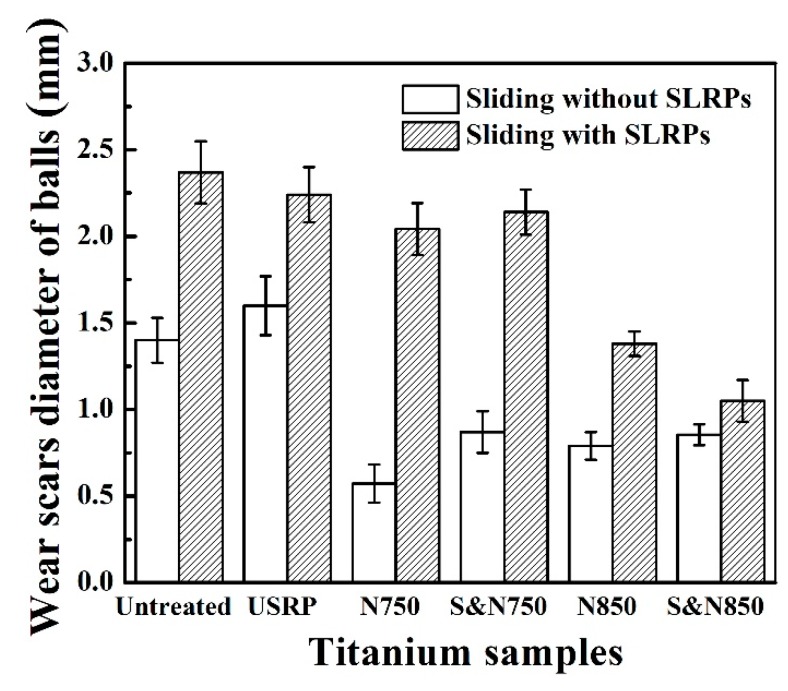
Wear scar diameter of the balls sliding against the Ti treated by USRP pre-treatment and plasma nitriding.

**Figure 12 materials-12-03260-f012:**
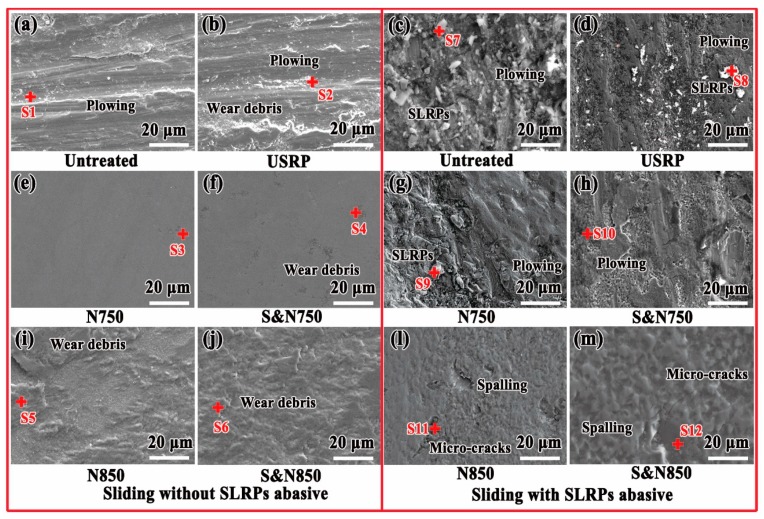
Typical secondary electron SEM image showing the surface morphology of the worn surface on the Ti treated by USRP pre-treatment and plasma nitriding sliding (**a**, **b**, **e**, **f**, **i** and **j**) without and (**c**, **d**, **g**, **h**, **l** and **m**) with abrasive SLRPs.

**Table 1 materials-12-03260-t001:** Nominal chemical composition of commercial TA2 (wt.%).

Elements	Ti	Fe	O	C	H	N	Si
Content	>99	≤0.30	≤0.35	≤0.08	≤0.015	≤0.03	≤0.15

**Table 2 materials-12-03260-t002:** Ultrasonic surface rolling processing (USRP) treatment parameters.

Vibration Frequency (kHz)	Amplitude (μm)	Load (N)	Spindle Speed (rpm)	Feed Rate (mm/rev)	Tip Diameter (mm)	Shots per mm^2^
20	30	350	200	0.05	15	35,340

**Table 3 materials-12-03260-t003:** Detailed parameters of plasma nitriding.

Voltage (V)	Current (A)	Pressure (Pa)	Temperature (℃)	Leakage Rate (Pa/h)	Duration (h)
−1100~−800	25~40	450~850	750 or 650	0.05	4

**Table 4 materials-12-03260-t004:** EDS analysis results show the elementary composition of wear debris on a worn surface.

Tests Condition	Samples	Spectrum No.	Element Content (wt.%)						
			Ti	N	Si	O	Al	Ca	Fe
Sliding without SLRPs abrasive	Untreated	S1	87	----	4	9	----	----	----
	USRP	S2	83	----	5	12	----	----	----
	N750	S3	57	4	18	20	----	----	----
	S&N750	S4	55	3	19	23	----	----	----
	N850	S5	40	2	25	32	----	----	----
	S&N850	S6	37	2	29	32	----	----	----
Sliding with SLRPs abrasive	Untreated	S7	18	----	9	46	7	10	9
	USRP	S8	16	----	8	43	7	10	16
	N750	S9	15	2	6	49	6	9	12
	S&N750	S10	13	2	6	48	8	11	12
	N850	S11	9	1	10	50	6	9	15
	S&N850	S12	8	1	11	49	7	11	12

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
