# Peer review of "Abrasive Wear Resistance of Plasma-Nitrided Ti Enhanced by Ultrasonic Surface Rolling Processing Pre-Treatment"

_materials, 2019, doi:10.3390/ma12193260_

Round 1
Reviewer 1 Report
Dear Authors,
The subject of the work is very important. The publication is fairly consistent, but requires slight correction:
- fig. 12, the description above the photos must be corrected as described below the figure;
- chapter Discussion could be described more clearly, also some information contained in this chapter was described earlier in the chapter Results. Both chapters could be one, e.g. Result and discussion.
Sincerely,
Reviewer
Author Response
Dear Editors,
I am the corresponding author of a paper (Manuscript ID: materials-588948) submitted online to Materials, which is titled with, Abrasive wear resistance of plasma-nitrided Ti enhanced by ultrasonic surface rolling processing pre-treatment
Here I write to you by detailed explanations about the paper when the revised version is submitted online. We carefully proofread the paper and the reviewers’ instruction, and based on them, any appropriate revisions were made to the paper. For details, see below:
Response: Thanks for your comment. According to your good suggestion, the description above Figure 12 has been corrected; and the chapter “Discussion” has been re-written and merged into the chapter “3. Result and discussion”. Please find those correction in revised Manuscript (Please see the attachment).

Reviewer 2 Report
Line 83
The title of this section should be modified, because it contains also other information i.e. XRD, SEM, EDS, 3-D imaging.
Line 88
The type of EDS system should be given and parameters of the analysis (voltage, etc.)
Line 90
Where in the article is any microphotograph of the cross-section?
Section 2.3. Tribo tests
Line 104
Please, include information about the producer of the tribotester.
Please, give information about the distance of the test (although it is possible to calculate it) and the diameter of the wear track.
Please, describe specifically the protocol of the wear coefficient determination. Was the wear of the counter-probe taken into account?
Section 3.1. Microstructure and morphology
In this section, microphotographs of the cross-sections of all types of the surface layers should be included.
Section 3.3.
Line 153. What is the friction coefficient of Ti? Friction coefficient is characteristic for the tribological system consisted of minimum 2 bodies.
In some cases, the friction coefficient is higher than 1, i.e. 1.2. This is very high value for the tribological system consisted of two hard materials, moreover different materials (metal and ceramics). Please, try to explain or comment it.
Section 3.4
Usually, the bigger friction coefficient leads to the bigger wear rate. In this manuscript, the situation is opposite, please try to comment it.
Table 4.
What was the area of the EDS analysis?
Presenting the results of the quantitative EDS results using 2 decimals is not proper.
Author Response
Dear Editors,
I am the corresponding author of a paper (Manuscript ID: materials-588948) submitted online to Materials, which is titled with, Abrasive wear resistance of plasma-nitrided Ti enhanced by ultrasonic surface rolling processing pre-treatment
Here I write to you by detailed explanations about the paper when the revised version is submitted online. We carefully proofread the paper and the reviewers’ instruction, and based on them, any appropriate revisions were made to the paper. For details, see below:
(Reviewer 1) The subject of the work is very important. The publication is fairly consistent, but requires slight correction:
- fig. 12, the description above the photos must be corrected as described below the figure;
- chapter Discussion could be described more clearly, also some information contained in this chapter was described earlier in the chapter Results. Both chapters could be one, e.g. Result and discussion.
Response 1: Thanks for your comment. According to your good suggestion, the description above Figure 12 has been corrected; and the chapter “Discussion” has been re-written and merged into the chapter “3. Result and discussion”. Please find those correction in revised Manuscript.
(Reviewer 2) The title of this section should be modified, because it contains also other information i.e. XRD, SEM, EDS, 3-D imaging. (Line 88).
Response 2: According to the comment, the title of the section has been modified as Microstructure and microhardness Characterization.
The type of EDS system should be given and parameters of the analysis (voltage, etc.) (Line 90)
Response 3: The parameters of the EDS analysis have been supplemented into the revised Manuscript.
Where in the article is any microphotograph of the cross-section?
Response 4: The microphotograph of the cross-section of the Ti treated by USRP and plasma nitriding have been supplemented as Figure 5 in the revised Manuscript. Please find it.
Section 2.3. Tribo tests, (Line 104) Please, include information about the producer of the tribotester.
Response 5: The parameters of the EDS analysis have been supplemented into the revised Manuscript.
Please, give information about the distance of the test (although it is possible to calculate it) and the diameter of the wear track.
Response 6: The diameter of wear track is 2.2 mm; and, the total sliding distance of tribo-tests is about 83.02 m.
Please, describe specifically the protocol of the wear coefficient determination. Was the wear of the counter-probe taken into account?
Response 7: After tribo-tests, wear volumes of the Ti slices measured by 3D profiler were converted to wear rate (K) according to the Archard wear equation:
K=V/(F•S)
where V is the wear volume, F is the normal load, S is the sliding distance, and K is the wear rate per unit load and per unit distance.
Wear volume of the counter-ball was shown via the diameter of wear scar which can be converted to wear rate via calculation.
Section 3.1. Microstructure and morphology, In this section, microphotographs of the cross-sections of all types of the surface layers should be included.
Response 8: The microphotograph of the cross-section of the Ti treated by USRP and plasma nitriding have been supplemented as Figure 5 in the revised Manuscript. Please find it.
Section 3.3. Line 153. What is the friction coefficient of Ti? Friction coefficient is characteristic for the tribological system consisted of minimum 2 bodies.
Response 9: Thanks for your comment. According to your suggestion, the description of “friction coefficient of Ti” has been corrected as “Friction coefficients of the Ti discs treated by USRP pre-treatment and plamsa nitriding sliding agianst SiO2 ball with SLRPs abasive”.
In some cases, the friction coefficient is higher than 1, i.e. 1.2. This is very high value for the tribological system consisted of two hard materials, moreover different materials (metal and ceramics). Please, try to explain or comment it.
Response 10: Under the condition of sliding without SLRPs abasive, the improvement of surface hardness via USRP and plasma nitrding is effective to enhance adhesive the wear resitance of the Ti, according to Archard relationship. Nevertheless, the adhesion between the ball and the Ti discs and the high surface roughness cause a friction coefficient higher than 1. Similar results also can be found in Ref [12] of the revised Manuscript.
Section 3.4 Usually, the bigger friction coefficient leads to the bigger wear rate. In this manuscript, the situation is opposite, please try to comment it.
Response 11: The hard SLRPs abrasive cause sever micro-cutting for the Ti with relatively low hardness, such as the untreated, USRP, N750 and S&N750 Ti. For the Ti plasma-nitrided under 850 ℃ with a high hardness, the anti-micro-cutting ability are enhanced. Accordingly, the N850 and S&N850 Ti show a relative low wear rate (Fig. 9 in revised Manuscript). Wear mechanism of the Ti plasma-nitrided under 850 ℃ is fatigue wear resulted from the rolling of the SLRPs abrasive; especially, the large-sized grain (Fig. 5 in revised Manuscript) caused by secondary recrystallization during plasma nitriding at high temperature always causes poor bearing capacity and the resistance of crack propagation and growth. Furthermore, SLRPs abrasive can impede the directly connect and adhesion between the ball and the Ti discs; and the co-action polishing and rolling of SLRPs abrasive finally results in a low friction coefficient, comparing with the condition of sliding without SLRPs abrasive. Under the condition of sliding without SLRPs abrasive, the improvement of surface hardness via USRP and plasma nitrding is effective to enhance adhesive the wear resistance of the Ti, according to Archard relationship. Nevertheless, the adhesion between the ball and the Ti discs and the high surface roughness cause a friction coefficient higher than 1 (Fig. 7 in revised Manuscript). As a result, high friction coefficient and low wear rate occurred comparing with the condition of sliding with SLRPs abrasive.
Table 4. What was the area of the EDS analysis?
Response 12: Thanks for your comment. The area of the EDS analysis has been marked in Figure 12 of the revised Manuscript.
Presenting the results of the quantitative EDS results using 2 decimals is not proper.
Response 13: Thanks for your comment. the results of the quantitative EDS results has been corrected in the revised Manuscript.

Reviewer 3 Report
Referee’s comments on the submited manuscript entitled :
Abrasive wear resistance of plasma-nitrided Ti enhanced by ultrasonic surface rolling processing pre-treatment
I/ General Comments
Abstract
It would have been better to first show the experimental abrasive wear results, indicating then their difference with ones obtained after other surface treatments and then discuss on the mechanisms proposed to explain such influence of SLRP and PN on the wear behaviour.
Experimental results, presentation discussion
It is strongly recommend to the authors to check the proposed version. Indeed, some Figures are not well referenced, more not introduced in the manuscript (see for example Figure 11 for which there is no related discussion text). In some cases, the Figure Caption is not in relation of the content of the Figure (Figure 6C for example). In some cases, the discussion text is not in agreement with the content of the Figure.
Annealing treatments have been achieved at different temperatures, 750°C and 850°C. Not the same experimental results have been provided allowing the full comparison between the all annealed samples. For example, there is no SEM image for annealing treatment at 750°C. Could the authors check this and provide all the experimental results allowing complete comparison between the various annealing treatment. If they consider that it is not necessary, justification has to be provided.It is requested also to check if the Figure captions as well as the text describing the Figure are in agreement with the content of the Figure itself (see for example Figure 6C, Figure 11).
Due to such noted discrepancies along the text, the discussion has to be checked and revised.
Presentation of Results and References
For references from Literature, the names of various publications are not well noted, as well as the right year or page number.Suggestions have been done by the Referee
II/ Specific Comments
Abstract
Page 1/ 12 / line 15
[...] For propose of investigating [...]
Referee’s comment and suggestion
Please check english (propose instead of purpose). Suggestion from referee
[...] The objectives of the given work is to investigate the abrasive wear behaviours..[...] [...]
Page 1/ 12 / line 19
[...] &PN [...]
Referee’s comment
[...] and PN [...]
Introduction
Page 1 / Line 39
[...] has been adopted [...]
Referee’s suggestion
[...] has been proposed [...], [...] has been introduced [...], [...] has been achieved [...]
Page 2 / Line 47
[...] Ge et al. [...]
to be modified as
[...] Ge et al. [...] (et al in italics)
Page 2 / Line 49
[...] Korshunov et al. [...]
to be modified as
[...] Korshunov et al. [...] (et al in italics, and space between korshunov and et)
Page 2 / Line 50
[...] She et al. [...]
to be modified as
[...] She et al. [...] (et al in italics)
Page 2/lines 52 and 53
[...] Unfortunately, abrasive wear behaviours of plasma-nitrided Ti pre-treated by USRP still far to be understood. [...]
Referee’s comment
The referee does not understand the use of the term « unfortunately ». Indeed, the influence of the USRP being positively demonstrated. A suggestion of the referee is to replace this word by « Nevertheless », indicating that even if the positive effect of USRP has been demonstrated, no mechanism has been proposed to explain such an improvement.
Experimental details
2.1. Materials and surface treatments
Page 2 / line 62
[...] Annealed commercial titanium TA2 [...]
Referee’s comment
Could the authors precise the name of the company providing the annealed commercial Titanium TA2.
Page 2 / line 63
[...] The chemical composition of TA2 [...]
Referee’s comment
Could the authors precise the origin of the chemical composition of TA2 (themselves, providing company) Could the authors precise how the composition has been determined (ICP – AES, else ?).
Page 2 / line 76
[...] After leakage rate test [...]
Referee’s comment
From the referee’s point of view, unless the authors precise why they add this precise point, this sentence is not necessary. Sure, that the basic organisation of experimental work has to be achieved in order to obtain right experimental results.
Page 3 / line 85
[...] 1.5406 Å [...]
to be replaced by
[...] 0.15406 nm [...] (Å is not a international standard unit)
Page 3 / Lines 89 and 90
[...] Electron Back- Scattered Diffraction, [...]
Referee’s comment
Electron Back- Scattered Diffraction is not describing an apparatus or a technique. It is the physics involved to achieve image. The apparatus is Scanning Electron Microscope
to be replaced by
[...] Scanning electron Microscope (electron backscattered image mode) [...]
Page 3 / Line 98 and Figure 1b
[...] A Gaussian distribution ranging from 10 to 150 μm [...]
Referee’s comment
As the size distribution is dependant on the lowest threshold used to measure the particle size, could the authors provide this information. Could the authors confirm that there is no particle size below 10 μm and that it is not an induced effect of the threshold.
Page 3 / Figure 1C
Referee’s comment
The FeO phase is identified based on a single X-Ray diffraction peak ; more, a single peak of very low intensity compared to others ones which have not been quoted. From the referee’s point of view, such a strong affirmation of the detection of the FeO phase is very doubtful.
Page 3 / Lines 102 and 103
[...] (a) morphology shown via SEM image, [...]
Referee’s comment
Could the authors precise the electron image mode used for obtaining the SEM image.
Page 4 / Line 138
[...] Figure 3. Typic SEM morphologies of the (a) untreated and (b) USRP Ti plasma-nitrided at 850 ℃. [...]
Referee’s comment
Could the authors precise the electron image mode used for obtaining the SEM image. As mentioned in the « General Comments from Referee », the given SEM image are corresponding to a 850°C treatment. There is not the equivalent observation for 750°C treatment. Please provide or justify why it is not necessar
Page 5/ line
[...] Figure 5. Microhardness of plasma-nitrided Ti pre-treated by USRP along the depth. [...]
Referee’s suggestion
to Replace depth by thickness.
Page 6 /Lines 163 and 164
[...] Figure 6. Friction coefficients of the (a) untreated and USRP pretreated, (b) N750 and S&750 and (c) 163 N750 and S&750 Ti sliding with and without SLRPs abasive [...]
Referee’s comment
There is a mystake in the caption corresponding to Figure 6C. The temperature noted in Figure is 850°C, not 750°C as reported in the caption.
Page 8 / Lines 204 and 205
[...] The diameter of wear scar on the ball sliding with the N850 Ti is much larger than that of the S&N850. [...]
Referee’s comment
If this sentence is related to Figure 11, the experimental results shown in Figure 11 are in contradiction with the sentence. In fact, from Figure 11, the wear scars diameters of ball for N850 Ti are slightly smaller (not much larger) than the ones of S&N850 Ti (both conditions sliding without or with SLRPs
Page 8 / Lines 206 to 208
Referee’s comment
Figure 11 is not introduced in the text. No explanation, no discussion. Just the Figure 11 and Figure 11 caption have provided. It is resquested to add text related to this Figure (maybe in lines 204 and 205 from Page 8) or to suppress this Figure.
Page 9 / Lines246 and 247
Figure 12. Typical SEM image showing the surface morphology of worn surface on the Ti treated by USRP pre-treatment and plasma nitriding sliding (a, b, e, f, i and j) without and (c, d, g, h, l and m) with SLRPs abasive
Referee’s comment
Could the authors precise the electron image mode used for obtaining the SEM image.
Page 9 / Line 249
Table 4. EDS analyze results shows the elementary composition of wear debris on worn surface.
Referee’s comment
Table 4. EDS analysis results shows the elementary composition of wear debris on worn surface.
III/ References
Page 10 / Lines 272 and 273
Zacny, K., Lunar Drilling, Excavation and Mining in Support of Science, Exploration, Construction, and In Situ Resource Utilization (ISRU). 2013.
to be replaced by
Zacny K. (2012) Lunar Drilling, Excavation and Mining in Support of Science, Exploration, Construction, and In Situ Resource Utilization (ISRU). In: Badescu V. (eds) Moon. Springer, Berlin, Heidelberg, pp 235 - 265 DOI : 10.1007/978-3-642-27969-0_10
Page 10 / Lines 274 to 276
Kobrick, R.; Budinski, K.; Street, K.; Klaus, Three-Body Abrasion Testing using Lunar Dust Simulants to Evaluate Surface System Materials, D. In, International Conference on Environmental Systems, 2010.
to be replaced by
NASA Technical Report (NASA/TM—2010-216781) prepared for 40th International Conference on Environmental Systems cosponsored by the AIAA, AICHE, ASME, SAE Environmental Systems Committee, and ICES International Committee, Barcelona, Spain, July 11–15, 2010; July 11, 2010 - July 15, 2010; Barcelona; Spain.
Page 10 / Lines 279 to 281
She, D.; Wen, Y.; Kang, J.; Fei, H.; Wang, C.; Jing, S.; Vacuum tribological properties of titanium enhanced via ultrasonic surface rolling processing pre treatment and plasma nitriding. Tribology Transaction 2017, (7), 00-00.
Referee’s comment, see : https://www.tandfonline.com/action/showCitFormats?doi=10.1080%2F10402004.2017.1380870)
to be replaced by
She, D.; Wen, Y.; Kang, J.; Fei, H.; Wang, C.; Jing, S.; Vacuum tribological 279 properties of titanium enhanced via ultrasonic surface rolling processing pretreatment and plasma nitriding. Tribology Transaction 2018, Volume 61 - Issue 4, Pages 612-620
Page 10 / Lines 282 and 283
Li, X.; Yue W.; Huang, F.; Kang, J.; Zhu, L.; Tian, B.; Tribological behaviour of 282 textured titanium under abrasive wear. Surface Engineering, 2018, 1-9
Referee’s comment, see : https://www.tandfonline.com/action/showCitFormats?doi=10.1080%2F02670844.2018.1512233
to be replaced by
Xingliang Li, Wen Yue, Fei Huang, Jiajie Kang, Lina Zhu & Bin Tian Tribological behaviour of textured titanium under abrasive wear, Surface Engineering, 2019, vol. 35, issue 4, 378-386
Page 10 / Line 286
Surface coatings & Technology
to be replaced by
Surface Coatings & Technolog
Page 10 / Lines 295 to 298
Hovsepian, P. E.; Ehiasarian, A. P.; Petrov, I. TiAlCN/VCN nanolayer coatings suitable for machining of Al and Ti alloys deposited by combined high power impulse magnetron sputtering/unbalanced magnetron sputtering. S. E.,2010, 26, (8), 610-614.
Referee’s comment, see
https://www.tandfonline.com/action/showCitFormats?doi=10.1179%2F026708408X336337
to be replaced by
E. Hovsepian, A. P. Ehiasarian & I. Petrov, TiAlCN/VCN nanolayer coatings suitable for machining of Al and Ti alloys deposited by combined high power impulse magnetron sputtering/unbalanced magnetron sputtering, Surface Engineering, 2010, 26:8, 610-614
Page 11/ Lines 302 to 304
Ramaseshan, R.; Jose, F.; Rajagopalan, S.; Dash, S.; Preferentially oriented electron beam deposited TiN thin films using focused jet of nitrogen gas. S.E. 2016, 32, (11), 834-839
Referee’s comment : see,
https://www.tandfonline.com/action/showCitFormats?doi=10.1080%2F02670844.2016.1159832
to be replaced by
R. Ramaseshan, Feby Jose, S. Rajagopalan & S. DashPreferentially oriented electron beam deposited TiN thin films using focused jet of nitrogen gas,Surface Engineering, 2016, 32:11, 834-839
Page 11 / Lines 312 to 315
Korshunov, L. G.; Chernenko, N. L. J. P. o. M.; Metallography, Effect of severe 312 plastic deformation on the structure, microhardness, and wear resistance of the 313 surface layer of titanium subjected to gas nitriding. The Physics of Metals and 314 Metallography 2014, 115, (10), 1027-1036.
to be replaced by
Korshunov, L. G.; Chernenko, N. L. J. P. o. M.; Metallography, Effect of severe 312 plastic deformation on the structure, microhardness, and wear resistance of the 313 surface layer of titanium subjected to gas nitriding. Physics of Metals and Metallography 2014, 115, (10), 1027-1036.
Page 11 / Line 319 to 321
Zhang, Y.; Chen, S.; Fei, Y. U.; Jian, L. I.; Gao, H. J. C.;, Experimental study of mechanical properties of Lunar soil simulant CAS–1 under low stress. J. o. R. M. Engineering, 2015, 34, (1), 174-181.
to be replaced by
Zhang, Y.; Chen, S.; Fei, Y. U.; Jian, L. I.; Gao, H. J. C., Experimental study of mechanical properties of Lunar soil simulant CAS–1 under low stress. Journal of Rock Mechanics and Engineering, 2015, 34, (1), 174-181.
Page 11 / Lines 322 and 323
Weiblen, P. W.; Gordon, K., Characteristics of a Simulant for Lunar Surface Materials. L. C. 1988, 652, (652), 254.
to be replaced by
Weiblen, P. W.; Gordon, K., Characteristics of a Simulant for Lunar Surface Materials. Proceeding of Conference Abstracts presented to the Second Conference on Lunar Bases and Space Activities of the 21st Century. Held in Houston, TX, April 5-7, 1988. Sponsored by NASA, the American Institute of Aeronautics and Astronautics, LPI, AGU, the American Nuclear Society, the American Society of Civil Engineers, the Space Studies Institute, and the National Space Society. LPI Contribution 652, published by the Lunar and Planetary Institute, 3303 Nasa Road 1, Houston, TX 77058, 1988, p.254
Page 11 / Lines 324 and 325
Zheng, Y.; Wang, S.; Ouyang, Z.; Zou, Y.; Liu, J.; Li, C.; Li, X.; Feng, J., CAS-1 lunar soil simulant. J. A. i. S. R. 2009, 43, (3), 448-454.
to be replaced by
Zheng, Y.; Wang, S.; Ouyang, Z.; Zou, Y.; Liu, J.; Li, C.; Li, X.; Feng, J., CAS-1 lunar soil simulant. Advances in Space Research 2009, 43, (3), 448-454.
Author Response
Dear Editors,
I am the corresponding author of a paper (Manuscript ID: materials-588948) submitted online to Materials, which is titled with, Abrasive wear resistance of plasma-nitrided Ti enhanced by ultrasonic surface rolling processing pre-treatment
Here I write to you by detailed explanations about the paper when the revised version is submitted online. We carefully proofread the paper and the reviewers’ instruction, and based on them, any appropriate revisions were made to the paper. For details, see below:
I/ General Comments: (Abstract) It would have been better to first show the experimental abrasive wear results, indicating then their difference with ones obtained after other surface treatments and then discuss on the mechanisms proposed to explain such influence of SLRP and PN on the wear behaviour.
Response 14: Thanks for your good suggestions. The description of experimental results in Abstract has been rewritten as following:
Experimental results show SLRPs causes severe abrasive wear on the Ti plasma-nitrided at 750 ℃ via the mechanism of micro-cutting. Due to the formation of a harder and thicker nitriding layer, abrasive wear resistance of the Ti plasma-nitrided at 850 ℃ are enhanced; and its wear mechanism is dominant via fatigue. USRP pre-treatment is effective to enhance abrasive wear resistance of plasma-nitrided Ti, due to the enhancement of hardness and thickness of the nitride layer. Nevertheless, SLRPs significantly decreases the friction coefficient of Ti treated by USRP pre-treatment and PN, resulting from the rolling of granular abrasives with a small size impedes the adhesion of worn surface. Furthermore, USRP pre-treatment also causes the formation of a dimpled surface with greater number of micropores which keeps wear debris during tribo-tests; and finally, the polishing and rolling of wear debris results in a low friction coefficient (about 0.5).
(Experimental results, presentation discussion) It is strongly recommended to the authors to check the proposed version. Indeed, some Figures are not well referenced, more not introduced in the manuscript (see for example Figure 11 for which there is no related discussion text). In some cases, the Figure Caption is not in relation of the content of the Figure (Figure 6C for example). In some cases, the discussion text is not in agreement with the content of the Figure.
Annealing treatments have been achieved at different temperatures, 750°C and 850°C. Not the same experimental results have been provided allowing the full comparison between the all annealed samples. For example, there is no SEM image for annealing treatment at 750°C. Could the authors check this and provide all the experimental results allowing complete comparison between the various annealing treatment. If they consider that it is not necessary, justification has to be provided. It is requested also to check if the Figure captions as well as the text describing the Figure are in agreement with the content of the Figure itself (see for example Figure 6C, Figure 11). Due to such noted discrepancies along the text, the discussion has to be checked and revised.
Response 15: We carefully proofread the paper, and based on your comment, the appropriate revisions were made to the paper to improve the quality of the Manuscripts. Figures are well referenced and introduced. The description of Figure 11 has been supplemented. The SEM image for annealing treatment at 750°C also has been supplemented into Figure 3. The caption of Figure 6C has been corrected. The discussion text has been re-written. Please find those in revised manuscript.
Presentation of Results and References: For references from Literature, the names of various publications are not well noted, as well as the right year or page number. Suggestions have been done by the Referee
Response 16: Thanks for your good suggestions. References have been detailed modified in revised Manuscipt.
II/ Specific Comments
Abstract: (Page 1/ 12 / line 15) [...] For propose of investigating [...] Referee’s comment and suggestion Please check english (propose instead of purpose). Suggestion from referee: [...] The objectives of the given work is to investigate the abrasive wear behaviours..[...] [...]
(Page 1/ 12 / line 19) [...] &PN [...] Referee’s comment: [...] and PN [...]
Introduction: (Page 1 / Line 39) [...] has been adopted [...] Referee’s suggestion: [...] has been proposed [...], [...] has been introduced [...], [...] has been achieved [...]
(Page 2 / Line 47) [...] Ge et al. [...] to be modified as [...] Ge et al. [...] (et al in italics)
(Page 2 / Line 49) [...] Korshunov et al. [...] to be modified as [...] Korshunov et al. [...] (et al in italics, and space between korshunov and et)
(Page 2 / Line 50) [...] She et al. [...] to be modified as [...] She et al. [...] (et al in italics)
(Page 2/lines 52 and 53) [...] Unfortunately, abrasive wear behaviours of plasma-nitrided Ti pre-treated by USRP still far to be understood. [...] Referee’s comment: The referee does not understand the use of the term « unfortunately ». Indeed, the influence of the USRP being positively demonstrated. A suggestion of the referee is to replace this word by « Nevertheless », indicating that even if the positive effect of USRP has been demonstrated, no mechanism has been proposed to explain such an improvement.
Response 16: Thanks for your good suggestions, and based on those, we have try our best to proofread and corrected to make sure the revised manuscript is free of grammatical, spelling, and formatting errors.
Experimental details:
2.1. Materials and surface treatments
(Page 2 / line 62) [...] Annealed commercial titanium TA2 [...] Referee’s comment: Could the authors precise the name of the company providing the annealed commercial Titanium TA2.
(Page 2 / line 63) [...] The chemical composition of TA2 [...] Referee’s comment: Could the authors precise the origin of the chemical composition of TA2 (themselves, providing company) Could the authors precise how the composition has been determined (ICP – AES, else ?).
Response 17: Annealed commercial titanium TA2 (ASTM Gr.3) bar made by Baoji Titanium Industry Co., Ltd. was adopted in our investigation. The chemical composition of TA2 slices provided by the company was listed in Table 1 in manuscript.
Page 2 / line 76 [...] After leakage rate test [...] Referee’s comment: From the referee’s point of view, unless the authors precise why they add this precise point, this sentence is not necessary. Sure, that the basic organisation of experimental work has to be achieved in order to obtain right experimental results.
Response 18: Thanks for you good suggestion, and based on it, the sentence has been deleted.
(Page 3 / line 85) [...] 1.5406 Å [...] to be replaced by [...] 0.15406 nm [...] (Å is not a international standard unit)
Response 19: Thanks for your good suggestion, and based on it, the Å has been corrected as nm.
(Page 3 / Lines 89 and 90) [...] Electron Back- Scattered Diffraction, [...] Referee’s comment: Electron Back- Scattered Diffraction is not describing an apparatus or a technique. It is the physics involved to achieve image. The apparatus is Scanning Electron Microscope to be replaced by [...] Scanning electron Microscope (electron backscattered image mode) [...]
Response 20: Thanks for your comment, the Electron Back- Scattered Diffraction results in this manuscript is similar with the ross-sectional micro-photographs. Therefore, we delete the repetitive results and description in revised manuscript.
(Page 3 / Line 98 and Figure 1b) [...] A Gaussian distribution ranging from 10 to 150 μm [...] Referee’s comment: As the size distribution is dependant on the lowest threshold used to measure the particle size, could the authors provide this information. Could the authors confirm that there is no particle size below 10 μm and that it is not an induced effect of the threshold.
Response 21: In reality, we cannot make sure that there is no particle size below 10 μm, but the probability of observed the particle size below 10 μm or above 150 is very small. Therefore, we corrected the description to “[...]the distribution of simulated lunar regolith particles, here, also shows as a Gaussian distribution cantered around 80 μm[...]”.
(Page 3 / Figure 1C) Referee’s comment: The FeO phase is identified based on a single X-Ray diffraction peak; more, a single peak of very low intensity compared to others ones which have not been quoted. From the referee’s point of view, such a strong affirmation of the detection of the FeO phase is very doubtful.
Response 22: Thanks for your comment, the XRD results in this manuscript has been corrected in revised manuscript, indeed, there is no FeO phase in SLRPs.
(Page 3 / Lines 102 and 103 ) [...] (a) morphology shown via SEM image, [...] Referee’s comment: Could the authors precise the electron image mode used for obtaining the SEM image.
Response 23: SEM images in this manuscript are obtained via secondary electron image system.
24.(Page 4 / Line 138)[...] Figure 3. Typic SEM morphologies of the (a) untreated and (b) USRP Ti plasma-nitrided at 850 ℃. [...] Referee’s comment: Could the authors precise the electron image mode used for obtaining the SEM image. As mentioned in the « General Comments from Referee », the given SEM image are corresponding to a 850°C treatment. There is not the equivalent observation for 750°C treatment. Please provide or justify why it is not necessary.
Response 24: SEM images in this manuscript are obtained via secondary electron image system. Typic SEM morphologies showing surface morphology of the untreated and USRP Ti plasma-nitrided at 750 ℃ has been supplemented into Figure3.
(Page 5/ line) [...] Figure 5. Microhardness of plasma-nitrided Ti pre-treated by USRP along the depth. [...] Referee’s suggestion: to Replace depth by thickness.
(Page 6 /Lines 163 and 164) [...] Figure 6. Friction coefficients of the (a) untreated and USRP pretreated, (b) N750 and S&750 and (c) N750 and S&750 Ti sliding with and without SLRPs abasive [...] Referee’s comment: There is a mystake in the caption corresponding to Figure 6C. The temperature noted in Figure is 850°C, not 750°C as reported in the caption.
Response 25: Thanks for your suggestions, and based on those, the mistakes has been corrected in revised manuscript.
(Page 8 / Lines 204 and 205) [...] The diameter of wear scar on the ball sliding with the N850 Ti is much larger than that of the S&N850. [...] Referee’s comment: If this sentence is related to Figure 11, the experimental results shown in Figure 11 are in contradiction with the sentence. In fact, from Figure 11, the wear scars diameters of ball for N850 Ti are slightly smaller (not much larger) than the ones of S&N850 Ti (both conditions sliding without or with SLRPs).
(Page 8 / Lines 206 to 208) Referee’s comment: Figure 11 is not introduced in the text. No explanation, no discussion. Just the Figure 11 and Figure 11 caption have provided. It is resquested to add text related to this Figure (maybe in lines 204 and 205 from Page 8) or to suppress this Figure.
Response 26: Wear scar on the balls sliding against the N850 Ti with SLRPs abasive (Fig 10g in revised manuscripts) contains two distinct surface morphology: the bright one is shown as some micro-plows and irregular edges (like the balls sliding against the N750 and S&N750 Ti); and the dark one is presenting as a flat with regular circle edge (like the balls sliding against the S&N850 Ti). The wear scars diameters of ball for N850 Ti are much larger than the ones of S&N850 Ti (both conditions sliding without or with SLRPs). We have remarked it in Fig 10 and replotted it in Fig. 11. In revised manuscripts, Figure 11 has been provided and well described.
(Page 9 / Lines246 and 247) Figure 12. Typical SEM image showing the surface morphology of worn surface on the Ti treated by USRP pre-treatment and plasma nitriding sliding (a, b, e, f, i and j) without and (c, d, g, h, l and m) with SLRPs abasive. Referee’s comment: Could the authors precise the electron image mode used for obtaining the SEM image.
Response 27: SEM images in this manuscript are all obtained via secondary electron image system.
(Page 9 / Line 249) Table 4. EDS analyze results shows the elementary composition of wear debris on worn surface. Referee’s comment: Table 4. EDS analysis results shows the elementary composition of wear debris on worn surface.
Response 28: Thanks for your suggestion, and based on it, the caption of Table 4 has been corrected.
III/ References
(Page 10 / Lines 272 and 273) Zacny, K., Lunar Drilling, Excavation and Mining in Support of Science, Exploration, Construction, and In Situ Resource Utilization (ISRU). 2013. to be replaced by
Zacny K. (2012) Lunar Drilling, Excavation and Mining in Support of Science, Exploration, Construction, and In Situ Resource Utilization (ISRU). In: Badescu V. (eds) Moon. Springer, Berlin, Heidelberg, pp 235 - 265 DOI: 10.1007/978-3-642-27969-0_10
(Page 10 / Lines 274 to 276) Kobrick, R.; Budinski, K.; Street, K.; Klaus, Three-Body Abrasion Testing using Lunar Dust Simulants to Evaluate Surface System Materials, D. In, International Conference on Environmental Systems, 2010. to be replaced by
NASA Technical Report (NASA/TM—2010-216781) prepared for 40th International Conference on Environmental Systems cosponsored by the AIAA, AICHE, ASME, SAE Environmental Systems Committee, and ICES International Committee, Barcelona, Spain, July 11–15, 2010; July 11, 2010 - July 15, 2010; Barcelona; Spain. https://ntrs.nasa.gov/archive/nasa/casi.ntrs.nasa.gov/20100033103.pdf
(Page 10 / Lines 279 to 281) She, D.; Wen, Y.; Kang, J.; Fei, H.; Wang, C.; Jing, S.; Vacuum tribological properties of titanium enhanced via ultrasonic surface rolling processing pre-treatment and plasma nitriding. Tribology Transaction 2017, (7), 00-00. Referee’s comment, see: https://www.tandfonline.com/action/showCitFormats?doi=10.1080%2F10402004.2017.1380870)
to be replaced by She, D.; Wen, Y.; Kang, J.; Fei, H.; Wang, C.; Jing, S.; Vacuum tribological 279 properties of titanium enhanced via ultrasonic surface rolling processing pretreatment and plasma nitriding. Tribology Transaction 2018, Volume 61 - Issue 4, Pages 612-620
(Page 10 / Lines 282 and 283) Li, X.; Yue W.; Huang, F.; Kang, J.; Zhu, L.; Tian, B.; Tribological behaviour of 282 textured titanium under abrasive wear. Surface Engineering, 2018, 1-9
Referee’s comment, see https://www.tandfonline.com/action/showCitFormats?doi=10.1080%2F02670844.2018.1512233
to be replaced by
Xingliang Li, Wen Yue, Fei Huang, Jiajie Kang, Lina Zhu & Bin Tian Tribological behaviour of textured titanium under abrasive wear, Surface Engineering, 2019, vol. 35, issue 4, 378-386
(Page 10 / Line 286) Surface coatings & Technology to be replaced by Surface Coatings & Technology
(Page 10 / Lines 295 to 298) Hovsepian, P. E.; Ehiasarian, A. P.; Petrov, I. TiAlCN/VCN nanolayer coatings suitable for machining of Al and Ti alloys deposited by combined high power impulse magnetron sputtering/unbalanced magnetron sputtering. S. E.,2010, 26, (8), 610-614.
Referee’s comment, see https://www.tandfonline.com/action/showCitFormats?doi=10.1179%2F026708408X336337
to be replaced by E. Hovsepian, A. P. Ehiasarian & I. Petrov, TiAlCN/VCN nanolayer coatings suitable for machining of Al and Ti alloys deposited by combined high power impulse magnetron sputtering/unbalanced magnetron sputtering, Surface Engineering, 2010, 26:8, 610-614
(Page 11/ Lines 302 to 304) Ramaseshan, R.; Jose, F.; Rajagopalan, S.; Dash, S.; Preferentially oriented electron beam deposited TiN thin films using focused jet of nitrogen gas. S.E. 2016, 32, (11), 834-839
Referee’s comment: see, https://www.tandfonline.com/action/showCitFormats?doi=10.1080%2F02670844.2016.1159832
to be replaced by R. Ramaseshan, Feby Jose, S. Rajagopalan & S. Dash. Preferentially oriented electron beam deposited TiN thin films using focused jet of nitrogen gas,Surface Engineering, 2016, 32:11, 834-839
(Page 11 / Lines 312 to 315) Korshunov, L. G.; Chernenko, N. L. J. P. o. M.; Metallography, Effect of severe 312 plastic deformation on the structure, microhardness, and wear resistance of the surface layer of titanium subjected to gas nitriding. The Physics of Metals and 314 Metallography 2014, 115, (10), 1027-1036.
to be replaced by Korshunov, L. G.; Chernenko, N. L. J. P. o. M.; Metallography, Effect of severe 312 plastic deformation on the structure, microhardness, and wear resistance of the surface layer of titanium subjected to gas nitriding. Physics of Metals and Metallography 2014, 115, (10), 1027-1036.
(Page 11 / Line 319 to 321) Zhang, Y.; Chen, S.; Fei, Y. U.; Jian, L. I.; Gao, H. J. C.;, Experimental study of mechanical properties of Lunar soil simulant CAS–1 under low stress. J. o. R. M. Engineering, 2015, 34, (1), 174-181. to be replaced by
Zhang, Y.; Chen, S.; Fei, Y. U.; Jian, L. I.; Gao, H. J. C., Experimental study of mechanical properties of Lunar soil simulant CAS–1 under low stress. Journal of Rock Mechanics and Engineering, 2015, 34, (1), 174-181.
(Page 11 / Lines 322 and 323) Weiblen, P. W.; Gordon, K., Characteristics of a Simulant for Lunar Surface Materials. L. C. 1988, 652, (652), 254.
to be replaced by
Weiblen, P. W.; Gordon, K., Characteristics of a Simulant for Lunar Surface Materials. Proceeding of Conference Abstracts presented to the Second Conference on Lunar Bases and Space Activities of the 21st Century. Held in Houston, TX, April 5-7, 1988. Sponsored by NASA, the American Institute of Aeronautics and Astronautics, LPI, AGU, the American Nuclear Society, the American Society of Civil Engineers, the Space Studies Institute, and the National Space Society. LPI Contribution 652, published by the Lunar and Planetary Institute, 3303 Nasa Road 1, Houston, TX 77058, 1988, p.254
(Page 11 / Lines 324 and 325) Zheng, Y.; Wang, S.; Ouyang, Z.; Zou, Y.; Liu, J.; Li, C.; Li, X.; Feng, J., CAS-1 lunar soil simulant. J. A. i. S. R. 2009, 43, (3), 448-454. to be replaced by
Zheng, Y.; Wang, S.; Ouyang, Z.; Zou, Y.; Liu, J.; Li, C.; Li, X.; Feng, J., CAS-1 lunar soil simulant. Advances in Space Research 2009, 43, (3), 448-454.
Response 16: Thanks very much for your good suggestions. References have been detailed modified in revised Manuscript.

Round 2
Reviewer 2 Report
There are some errors in some words (for example "agianst"). The text should be carefully checked.
Author Response
Reviewer’s comment: There are some errors in some words (for example "agianst"). The text should be carefully checked.
Response: Thanks for your comment. We have carefully proofread the manuscript; and try our best to correct all grammatical and spelling errors. For details, see in revised manuscript.
Reviewer 3 Report
Referee’s comments on the submited manuscript (Revised Version) entitled :
Abrasive wear resistance of plasma-nitrided Ti enhanced by ultrasonic surface rolling processing pre-treatment
General Comments
Nanostructure induced by treatment :
Many times, the authors speak on nanocrystal, nano-particles. As such structural evolution induced by treatment is very important, it is requested to provide strong arguments to support the presence of nanostructured materials. Then, clear determination of the XRD peak width have to be provided, as well as the method to calculate the grain size including the instrumental contribution width to the XRD peaks.
XRD patterns :
Concerning XRD patterns, the referee has done this following comment (The FeO phase is identified based on a single X-Ray diffraction peak ; more, a single peak of very low intensity compared to others ones which have not been quoted. From the referee’s point of view, such a strong affirmation of the detection of the FeO phase is very doubtful.)
Clearly, the authors, in the revised version, do take into account this comment by suppressing the FeO Phase from the identified set ones. Nevertheless, it would have been important to directly answer to the referee by explaining what happen with this peak formerly attributed to this FeO Phase. Is it now unattributed, is it attributed to another phase, which one ?
More, comparing the XRD peaks analysis between the initially provided manuscript and the revised one, not only the FeO has disappeared, but also Al2O3. New phases have been identified, Al2SiO5, CaCO3 and Cr2O3.
With so large differences between informations provided between two versions of manuscript, it is normally expected to have specific comment from authors addressed to reviewers, explaining the origin of such differences (new XRD pattern acquisition, new XRD analyses ?).
How the authors could explain also the two versions of the text commenting the given XRD patterns
Initial text
[...] According to the characteristics of lunar mare basalts, simulated lunar regolith particles composed of SiO2, Al2O3, CaO, TiO2 and FeO were made from volcanic [22, 23]. [...]
Revised version – Page « / Lines 104 and 105 :
[...] According to the characteristics of lunar mare basalts, simulated lunar regolith particles composed of SiO2, Al2SiO5, CaCO3, Fe2O3 and Cr2O3 were made from volcanic [22, 23]. [...]
It is strange from the referee’s point of view to see Al203, CaO and FeO as well as TiO2 in the first version, no more in the revised one, being replaced by Al2SiO5, CaCO3 and Fe2O3, without mention on the disappearance of any phase containing Ti atoms.
Based on the own reading by referee of reference [22], [22] has been found to mention SiO2, Al2O3, CaO, FeO, MgO and TiO2. There is no mention of Al2SiO5, CaCO3, Fe2O3, Cr2O3.
(see : http://adsabs.harvard.edu/full/1988LPICo.652..254W)
Based on the own reading by referee of reference [23], [23] has been found to mention SiO2, TiO2, Al2O3, FeO, MnO, MgO, CaO, Na2O, K2O, P2O5. There is no mention of Al2SiO5, CaCO3, Fe2O3, Cr2O3
It looks as if the authors interpretate the previously published results [22, 23] as a function of their own results, which are noted to be different between initial and revised version of the same manuscript.It is clearly requested from authors to provide explanation to referee and to provide the true information from the references 22 and 23.
Specific Comments :
Page 2 / Line 70
Referee’s comment
The composition is expressed in wt. % in Table 1. In Table 4 (Page 11 / Line 301), the compositions are expressed in at. %. It is requested to provided informations on the composition in the same units. Without using the same unit, it is very difficult to see the influence of the treatment on the composition.
Page 3 / 13 – Line 86
[...] 2.2. Microstructure and microhardness Characterization [...]
to be replaced by
[...] 2.2. Microstructure and microhardness characterization [...]
Page 3 / 13 – Line 103
[...] also shows as a Gaussian distribution cantered around 80 μm [...]
to be replaced by
[...] also shows as a Gaussian distribution centered around 80 μm [...]
Page 4 / 13 – Line 131 - 132
[...] Diffraction peaks of the a-Ti phase for the USRP Ti are broader and weaker than that for the untreated Ti, [...]
Referee’s comment
As it seems very important for the authors to report on the nanocrystals, nanoparticles formation induced by treatment, strong arguments are requested. Therefore, it is requested to provide a Table indicating clearly the XRD peak widths for each phase (width in 2 theta, grain size calculation), before and after treatment. It is requested to provide in complement to this table, the mathematical equation used to obtain the XRD peak widthness substracting the instrumental XRD contribution and just taking into account the contribution from grain size refinement.
If the mentioned references [6, 7] are issued from the same research consortium and on the same materials treated in the same conditions, instead of providing the proposed complementary Table, authors have to clearly mentioned this feature (same materials, same treatment in references 6 and 7).
Page 4 / 13 – Line 135
[...] including a-Ti(N), [...]
to be replaced by
[...] including a-Ti(N), [...]
(Referee : a to be replaced by alpha, symbol letter)
Page 4 / Line 141
[...] Fig. 3 show typic SEM surface morphologies [...]
to be replaced by
[...] Fig. 3 show typical SEM surface morphologies [...]
Page 4 / Lines 144 and 145
[...] nevertheless, the N750 and N850 show [...]
to be replaced by
[...] nevertheless, the N750 and N850 Ti show [...]
Page 4 / Lines 145 and 146
[...] Obviously, USRP pre-treatment results in the formation of small-sized nitride nano-particles. [...]
Referee’s comment :
Such a formation of small-sized nitride nano-particles is not so obvious from the referee’s point of view.
Furthermore, the sentence in itself is not clear ? small-sized for nano-particles ? what does it means ??
Page 5 / Line 155
[...] Figure 3. Typic SEM images showing surface morphologies[...]
to be changed by
[...] Figure 3. Typical SEM images showing surface morphologies [...]
Referee’s comment
It is requested to provide the information on the SEM image mode (secondary electron image, backscattered one, else ?)
Page 5 / Line 161
[...] A gradient deformation layer with ultra-fine grain can be observed after USRP treatment. [...]
Referee’s comment
Could the authors introduced arrows or anything else to show on the Figure, the discussed area ? The referee does not undertand the sentence « gradient deformation » ? deformation of what ? The authors mention « nanocrystals », « nano-particles », « ultrafine grain » for describing the treated surface. Is there any difference hidden behind these distinct words, or do they mean the same observed features ? If it is the same feature, it would be better to use only one same sentence.
Page 5 / Lines 162 and 163
[...] Evidently, USRP pre-treatment is effective to enhance the thickness of the nitride layer comparing with the Ti directly treated by plasma nitriding, [...]
Referee’s comment
Could the authors introduced arrows or anything else to show on the Figure, the discussed area ? How the authors support the idea of a nitride layer ? Do they have any chemical analyses performed on the cross-section ?
Pages 5 / Lines 163 and 164 and Page 6/ Line 165
[...] which results from that defects in deformation layer caused by USRP pre-treatment can act as channel for atoms diffusion; and hence, promote the diffusion of nitrogen during nitriding [7, 16, 17, 20 and 25] [...]
Referee’s comment :
In fact, the authors, based on the published litterature, would like to explain the observed feature, so called « enhance the thickness of the nitride layer. The referee would like to suggest to modify the sentence as following
[...] Based on previously published results [7, 16, 17, 20 and 25], such a thickness enhancement has been proposed to be dued to.... [...]
Page 6 / Line 170
[...] Figure 5 [...]
Referee’s comment
Could the authors comment on the various red line located in the figure 5b, c, d, and e. From the referee’s point of view, it seems that the scale is different for the various Figures 5a, b, c, d and e. Could the authors provide this information on scale for each Figure.
Page 6 / Line 173
[...] Fig. 5 shows the micro hardness [...]
to be replaced by
[...] Fig. 6 shows the micro hardness [...]
Page 7 / Line 183, Line 187, Line 190, Line 194
[...] agianst [...]
to be replaced by
[...] against [...]
Page 7 / Line 194
[...] SiO2 [...]
to be replaced by
[...] SiO2 [...]
Page 7 / Line 196
[...] Typic 3D worn surface morphologies [...]
to be replaced by
[...] Typical 3D worn surface morphologies [...]
Page 7 / Line 209
[...] Figure 8. Typic 3D worn surface morphologies [...]
to be replaced by
[...] Figure 8. Typical 3D worn surface morphologies [...]
Page 9 / Line 232
[...] SLRPs abasive is effective [...]
to be replaced by
[...] SLRPs abrasive is effective [...]
Page 9 / Line 243
[...] Figure 11. wear scar diameter [...]
to be replaced by
[...] Figure 11. Wear scar diameter [...]
Page 9 / Line 243
[...] This reuslts is resulted from [...]
to be replaced by
[...] This results is resulted from [...]
Page 9 / Lines 262 and 263
[...] shown in Table 4, Wear debris [...]
to be replaced by
[...] shown in Table 4, Wear debris [...]
Page 9 / Line 265
[...] The reasons for the aggregation is attributed to the higher hardness [...]
Referee’s comment
[...] The referee does not understand the term « aggregation » used here. It is may be « degradation» ? the right term ? [...]
Page 9 / Line 267
[...] The hard SLRPs abrasive cause sever micro-cutting [...]
to be replaced by
[...] The hard SLRPs abrasive cause severe micro-cutting [...]
Page 10 / Line 283
[...] According to Archard relationship, the [...]
Referee’s comment
Please introduce a reference for the relationship allowing to get the related information on such a relationship.
Page 11 / Line 301
Referee’s comments
In Table 4 (Page 11 / Line 301), the compositions are expressed in at. %. The composition is expressed in wt. % in Table 1. It is requested to provided informations on the composition in the same units. Without using the same unit, it is very difficult to see the influence of the treatment on the composition. In order to be able to compare the informations provided in the both Tables 1 and 4, it would be helpful to have a comment from authors on the possible influence of the techniques used to perform such chemical analyses (sensitivity to light elements for example, and so on). Indeed, the information of the chemical technique used to provide chemical composition of the provided ingolt has not been given (yet asked/requested by referee from the first version of the manuscript).
Page 11 / Line 329
To be replaced by (suppressing the underlining or the weblink).
Author Response
Dear reviewer,
Thanks for your constructive comments which greatly enhances the quality of this manuscript. Here I write to you by detailed explanations about the paper when the revised version is submitted online. We carefully proofread the paper and the reviewers’ instruction, and based on them, any appropriate revisions were made to the paper. For details, see below:
General Comments:
(Nanostructure induced by treatment) Many times, the authors speak on nanocrystal, nano-particles. As such structural evolution induced by treatment is very important, it is requested to provide strong arguments to support the presence of nanostructured materials. Then, clear determination of the XRD peak width have to be provided, as well as the method to calculate the grain size including the instrumental contribution width to the XRD peaks.
Response 1: Thanks for your good comment. We cannot agree more that structural evolution induced by treatment is very important, and strong arguments for structural evolution like nanocrystallization should be provided. Nevertheless, such structural evolutions have been detailedly introduced in our previous works (Refence 6, 7, 9, and 20); and the materials, USRP pre-treatment condition and process of plasma nitriding in this paper is the same with that in Refence 6, 7 and 9. Especially, the main themes we reported here is the abrasive wear behaviors. Therefore, a part of arguments to support the presence of nanostructured materials, nano-particles, etc. are not described, in order to avoid repetition.
XRD patterns:
Concerning XRD patterns, the referee has done this following comment (The FeO phase is identified based on a single X-Ray diffraction peak; more, a single peak of very low intensity compared to others ones which have not been quoted. From the referee’s point of view, such a strong affirmation of the detection of the FeO phase is very doubtful.)
Clearly, the authors, in the revised version, do take into account this comment by suppressing the FeO Phase from the identified set ones. Nevertheless, it would have been important to directly answer to the referee by explaining what happen with this peak formerly attributed to this FeO Phase. Is it now unattributed, is it attributed to another phase, which one?
More, comparing the XRD peaks analysis between the initially provided manuscript and the revised one, not only the FeO has disappeared, but also Al2O3. New phases have been identified, Al2SiO5, CaCO3 and Cr2O3.
With so large differences between informations provided between two versions of manuscript, it is normally expected to have specific comment from authors addressed to reviewers, explaining the origin of such differences (new XRD pattern acquisition, new XRD analyses?).
How the authors could explain also the two versions of the text commenting the given XRD patterns
Initial text
[...] According to the characteristics of lunar mare basalts, simulated lunar regolith particles composed of SiO2, Al2O3, CaO, TiO2 and FeO were made from volcanic [22, 23]. [...]
Revised version – Page « / Lines 104 and 105 :
[...] According to the characteristics of lunar mare basalts, simulated lunar regolith particles composed of SiO2, Al2SiO5, CaCO3, Fe2O3 and Cr2O3 were made from volcanic [22, 23]. [...]
It is strange from the referee’s point of view to see Al203, CaO and FeO as well as TiO2 in the first version, no more in the revised one, being replaced by Al2SiO5, CaCO3 and Fe2O3, without mention on the disappearance of any phase containing Ti atoms.
Based on the own reading by referee of reference [22], [22] has been found to mention SiO2, Al2O3, CaO, FeO, MgO and TiO2. There is no mention of Al2SiO5, CaCO3, Fe2O3, Cr2O3.
(see : http://adsabs.harvard.edu/full/1988LPICo.652..254W)
Based on the own reading by referee of reference [23], [23] has been found to mention SiO2, TiO2, Al2O3, FeO, MnO, MgO, CaO, Na2O, K2O, P2O5. There is no mention of Al2SiO5, CaCO3, Fe2O3, Cr2O3
It looks as if the authors interpretate the previously published results [22, 23] as a function of their own results, which are noted to be different between initial and revised version of the same manuscript. It is clearly requested from authors to provide explanation to referee and to provide the true information from the references 22 and 23.
Response 2: Thanks for your good comment. The X-Ray diffraction, here, is very difficult to identify, resulting from the big noise background caused by SLRPs with a wide dimensional distribution. Therefore, there are many mistakes in the previous versions of manuscript. We are so sorry about that. Accordingly, we conducted the X-Ray diffraction again and replaced the previous version. Please find it in revised manuscript.
Specific Comments: (Page 2 / Line 70) Referee’s comment: The composition is expressed in wt. % in Table 1. In Table 4 (Page 11 / Line 301), the compositions are expressed in at. %. It is requested to provided informations on the composition in the same units. Without using the same unit, it is very difficult to see the influence of the treatment on the composition.
Response 3: Thanks for your good comment. The compositions in Table 1 and 4 are all expressed in wt. % in the revised manuscript.
4.(Page 3 / 13 – Line 86) [...] 2.2. Microstructure and microhardness Characterization [...] to be replaced by [...] 2.2. Microstructure and microhardness characterization [...]
(Page 3 / 13 – Line 103) [...] also shows as a Gaussian distribution cantered around 80 μm [...] to be replaced by [...] also shows as a Gaussian distribution centered around 80 μm [...]
Response 4: Thanks for your good suggestions. The faults in spelling and grammar has been corrected in the revised manuscript.
(Page 4 / 13 – Line 131 – 132) [...] Diffraction peaks of the a-Ti phase for the USRP Ti are broader and weaker than that for the untreated Ti, [...] Referee’s comment
As it seems very important for the authors to report on the nanocrystals, nanoparticles formation induced by treatment, strong arguments are requested. Therefore, it is requested to provide a Table indicating clearly the XRD peak widths for each phase (width in 2 theta, grain size calculation), before and after treatment. It is requested to provide in complement to this table, the mathematical equation used to obtain the XRD peak widthness substracting the instrumental XRD contribution and just taking into account the contribution from grain size refinement.
If the mentioned references [6, 7] are issued from the same research consortium and on the same materials treated in the same conditions, instead of providing the proposed complementary Table, authors have to clearly mentioned this feature (same materials, same treatment in references 6 and 7).
Response 5: Thanks for your good comment. We cannot agree more that grain size calculation should be provided. Nevertheless, such structural evolutions have been detailedly introduced in our previous works (Refence 6, 7, 9, and 20); and the materials, USRP pre-treatment condition and process of plasma nitriding in this paper is the same with that in Refence 6, 7 and 9. Especially, the main themes we reported here is the abrasive wear behaviors. Therefore, a part of arguments to support the presence of nanostructured materials are not described, in order to avoid repetition.
(Page 4 / 13 – Line 135) [...] including a-Ti(N), [...] to be replaced by [...] including a-Ti(N), [...] (Referee: a to be replaced by alpha, symbol letter)
(Page 4 / Line 141) [...] Fig. 3 show typic SEM surface morphologies [...] to be replaced by [...] Fig. 3 show typical SEM surface morphologies [...]
(Page 4 / Lines 144 and 145) [...] nevertheless, the N750 and N850 show [...] to be replaced by [...] nevertheless, the N750 and N850 Ti show [...]
Response 6: Thanks for your good suggestions. The faults in spelling and grammar has been corrected in the revised manuscript.
(Page 4 / Lines 145 and 146) [...] Obviously, USRP pre-treatment results in the formation of small-sized nitride nano-particles. [...] Referee’s comment:
Such a formation of small-sized nitride nano-particles is not so obvious from the referee’s point of view.
Furthermore, the sentence in itself is not clear? small-sized for nano-particles? what does it means??
Response 7: Thanks for your good comment. The sentence has been corrected as “[...] USRP pre-treatment results in the formation of nano-sized nitride particles. [...]”
(Page 5 / Line 155) [...] Figure 3. Typic SEM images showing surface morphologies[...] to be changed by [...] Figure 3. Typical SEM images showing surface morphologies [...]; Referee’s comment
It is requested to provide the information on the SEM image mode (secondary electron image, backscattered one, else?)
Response 8: Thanks for your good comment. The caption of Figure 3 has been corrected as “[...] Typical secondary electron SEM images showing surface morphologies. [...]”
(Page 5 / Line 161) [...] A gradient deformation layer with ultra-fine grain can be observed after USRP treatment. [...] Referee’s comment:
Could the authors introduced arrows or anything else to show on the Figure, the discussed area? The referee does not undertand the sentence « gradient deformation »? deformation of what? The authors mention « nanocrystals », « nano-particles », « ultrafine grain » for describing the treated surface. Is there any difference hidden behind these distinct words, or do they mean the same observed features? If it is the same feature, it would be better to use only one same sentence.
Response 9: Thanks for your good comment. We marked the discussed area like gradient deformation layer and nanocrystals layer on the figures, please find it on the Figure 5. « nanocrystals », « nano-particles », « ultrafine grain » did not mean the same. Nanocrystals is the nano-sized grains. Ultrafine grain, here, means the grains have been refined to a relatively small size comparing with the untreated Ti. Nano-particles is nano-sized particle, like nano-sized nitride particle, nano-sized SLRPs, nano-sized wear debris.
(Page 5 / Lines 162 and 163) [...] Evidently, USRP pre-treatment is effective to enhance the thickness of the nitride layer comparing with the Ti directly treated by plasma nitriding, [...] Referee’s comment:
Could the authors introduced arrows or anything else to show on the Figure, the discussed area? How the authors support the idea of a nitride layer? Do they have any chemical analyses performed on the cross-section?
Response 10: Thanks for your comment. The discussed area has been marked on the figures. In our opinions, the formation of nitriding layer with typical white-bright layer and diffusion layer is empirical after plasma nitriding, and hence we didn’t conduct any chemical analyses, here. Especially, pervious XRD results has demonstrated that the formation of nitride phase on the surface of the nitridied Ti.
(Pages 5 / Lines 163 and 164 and Page 6/ Line 165) [...] which results from that defects in deformation layer caused by USRP pre-treatment can act as channel for atoms diffusion; and hence, promote the diffusion of nitrogen during nitriding [7, 16, 17, 20 and 25] [...] Referee’s comment : In fact, the authors, based on the published litterature, would like to explain the observed feature, so called « enhance the thickness of the nitride layer. The referee would like to suggest to modify the sentence as following [...] Based on previously published results [7, 16, 17, 20 and 25], such a thickness enhancement has been proposed to be dued to.... [...]
Response 11: Thanks for your comment. It has been revised according to your good suggestion.
(Page 6 / Line 170) [...] Figure 5 [...] Referee’s comment: Could the authors comment on the various red line located in the figure 5b, c, d, and e. From the referee’s point of view, it seems that the scale is different for the various Figures 5a, b, c, d and e. Could the authors provide this information on scale for each Figure.
Response 12: Thanks for your comment. The discussed area and comments have been marked on the figures. Scale bar is the same for Figures 5a, b, c, d and e; and it has been drawn on each figure. The reasons for why the grains on the Ti treated by USRP and plasma nitriding show various sized, please find it in our previous work (REF [6, 7 and 9] in manuscript).
13 (Page 6 / Line 173) [...] Fig. 5 shows the micro hardness [...] to be replaced by [...] Fig. 6 shows the micro hardness [...]
(Page 7 / Line 183, Line 187, Line 190, Line 194) [...] agianst [...] to be replaced by [...] against [...]
( Page 7 / Line 194) [...] SiO2 [...] to be replaced by [...] SiO2 [...]
(Page 7 / Line 196) [...] Typic 3D worn surface morphologies [...] to be replaced by [...] Typical 3D worn surface morphologies [...]
(Page 7 / Line 209) [...] Figure 8. Typic 3D worn surface morphologies [...] to be replaced by [...] Figure 8. Typical 3D worn surface morphologies [...]
(Page 9 / Line 232) [...] SLRPs abasive is effective [...] to be replaced by [...] SLRPs abrasive is effective [...]
(Page 9 / Line 243) [...] Figure 11. wear scar diameter [...] to be replaced by [...] Figure 11. Wear scar diameter [...]
(Page 9 / Line 243) [...] This reuslts is resulted from [...] to be replaced by [...] This results is resulted from [...]
(Page 9 / Lines 262 and 263) [...] shown in Table 4, Wear debris [...] to be replaced by [...] shown in Table 4, Wear debris [...]
Response 13: Thanks for your good suggestions. The faults in spelling and grammar has been corrected in the revised manuscript.
(Page 9 / Line 265) [...] The reasons for the aggregation is attributed to the higher hardness [...] Referee’s comment: [...] The referee does not understand the term « aggregation » used here. It is may be « degradation» ? the right term ? [...]
Response 14: This sentence has been re-written as “The reasons for the aggregation of wear debris on the worn surface resulted from that the higher hardness (Fig. 6) caused by USRP pre-treatment can result in a higher wear rate of the ball.
15. (Page 9 / Line 267) [...] The hard SLRPs abrasive cause sever micro-cutting [...] to be replaced by [...] The hard SLRPs abrasive cause severe micro-cutting [...]
Response 15: Thanks for your good suggestions. The faults in spelling and grammar has been corrected in the revised manuscript.
(Page 10 / Line 283) [...] According to Archard relationship, the [...] Referee’s comment: Please introduce a reference for the relationship allowing to get the related information on such a relationship.
Response 16: Reference has been supplemented, here, in the revised manuscript.
(Page 11 / Line 301) Referee’s comments: In Table 4 (Page 11 / Line 301), the compositions are expressed in at. %. The composition is expressed in wt. % in Table 1. It is requested to provided informations on the composition in the same units. Without using the same unit, it is very difficult to see the influence of the treatment on the composition. In order to be able to compare the informations provided in the both Tables 1 and 4, it would be helpful to have a comment from authors on the possible influence of the techniques used to perform such chemical analyses (sensitivity to light elements for example, and so on). Indeed, the information of the chemical technique used to provide chemical composition of the provided ingolt has not been given (yet asked/requested by referee from the first version of the manuscript).
Response 17: Thanks for good comment. The compositions are all expressed in wt. % in the revised manuscript. We have supplemented that “During EDS elemental composition measurements, the accelerating voltage is 15 kV; and working distance is about 15 mm.”; and we don’t know what is the important parameter, you think, we must provide for EDS analyzes, please tell us.
(Page 11 / Line 329) https:// ntrs.nasa.gov/archive/nasa/casi.ntrs.nasa.gov/ 20100033103.pdf
To be replaced by (suppressing the underlining or the weblink).
Response 18: Thanks for your suggestion. It has been corrected in the revised manuscript.
Round 3
Reviewer 3 Report
The reviewer thanks the author for taking into account the previous comments on the V2 version.
For the V3 version, some remaining ckecks have to be performed.
Page 7/13 - Line 215
[...] two distinct surface morphology [...]
to be replaced by
[...] two distinct surface morphologies [...]
Page 8/13 – Line 237
[...] the dirctly plasma-nitrided Ti [...]
to be replaced by
[...] the directly plasma-nitrided Ti [...]
Page 8/13 – Line 237
[...] the Ti tretated by USRP [...]
to be replaced by
[...] the Ti treated by USRP [...]
Page 9/13 – Line 238
[...] exhibits a consistent size [...]
to be replaced by
[...] exhibiting a consistent size [...]
Page 9/13 – Line 238
[...] It’s worth noting that the SLRPs abasive resulted [...]
to be replaced by
[...] It’s worth noting that the SLRPs abrasive resulted [...]
Page 9/13 – Line 249
[...] Accodingly, the wear mechanism of the untreated [...]
to be replaced by
[...] Accordingly, the wear mechanism of the untreated [...]
Page 9/13 – Line 249
[...] is abasive wear [...]
to be replaced by
[...] is abrasive wear [...]
Page 9/13 – Line 258
[...] Some scaly wear debris are covered on the worn sueface of the N850 and [...]
to be replaced by
[...] Some scaly wear debris are covered on the worn surface of the N850 and [...]
Page 12/13 – Line 349
[...] Surface Engineering, 2010, 26, (8), 610-614. [...]
to be replaced by
[...] Surface Engineering, 2010, 26, (8), 610-614. [...]
Page 12/13 – Line 381
[...] Diamond Sliding Against Diferent Mating [...]
to be replaced by
[...] Diamond Sliding Against Different Mating [...]
Referee’s comment :
Please check « Fig » be written as « Fig. ». In many parts, the « . » is missing after « Fig »
(e.g. see page 3 – Line 101, Line 103, Page 7, Line 213, Line 215)